# Towards Worst-Case Guarantees with Scale-Aware Interpretability

## Abstract

Neural networks organize information according to the hierarchical, multi-scale structure of natural data. Methods to interpret model internals should be similarly scale-aware, explicitly tracking how features compose across resolutions and guaranteeing bounds on the influence of fine-grained structure that is discarded as irrelevant noise. We posit that the renormalisation framework from physics can meet this need by offering technical tools that can overcome limitations of current methods. Moreover, relevant work from adjacent fields has now matured to a point where scattered research threads can be synthesized into practical, theory-informed tools. To combine these threads in an AI safety context, we propose a unifying research agenda – called *scale-aware interpretability* – to develop formal machinery and interpretability tools that have robustness and faithfulness properties supported by statistical physics.

## 1 Introduction

By probing the internal structure of neural networks (NNs), the growing field of AI interpretability aims to improve our understanding, trust, and ability to audit AI systems. The health of the field relies heavily on tools capable of opening the black box to discover the mechanisms and patterns that govern AI behavior. Importantly, these tools – though engineering artefacts – should be bolstered by theoretical and empirical support. Sparse auto-encoders (SAEs), for example, grew out of a theoretical hypothesis about feature representations in NNs that was empirically verified in a toy setting (Elhage et al., 2022). This trajectory – from theory and experiment to scalable instrumentation – offers an attractive scientific pipeline for ambitious interpretability upon which we hope to iterate[1].

We argue that the renormalisation framework from statistical physics is particularly suited to the needs of ambitious interpretability, capable of i) describing the large amount of regular structure that NNs learn from data and ii) transforming those insights into tools with better robustness and faithfulness properties than the state of the art. Moreover, we contend that the field is now ready for coalescence, and call for coordination across the physics, neuroscience, CS, and AI safety communities to achieve the goals set out in this paper. Concretely, renormalisation formalizes three key aspects of NNs that current methods handle poorly: the importance of scale (granularity, resolution), relevance (which degrees of freedom matter at a particular scale), and coarse-graining (how irrelevant degrees of freedom are systematically ignored). We sketch the necessary background in Section 2. In section 3, we outline an interpretability framework that:

1. Finds natural scales in NNs for coarse-graining features during training and inference.

2. Extracts a principled, scale-dependent notion of feature relevance.

3. Leads to robust guarantees for when fine-grained fluctuations can be ignored at a chosen level of abstraction.

Note that 'renormalisation' is not a single prescriptive procedure, but a general framework for understanding how physical theories depend on scale. It has evolved to include a variety of techniques capable of making reliable predictions across contexts, from quantum materials to collider physics. We aim to build a similarly adaptable framework in NNs, an approach we're calling *Scale-Aware*

---

[1]For more detail about this pipeline and the discovery of SAEs, see Appendix B)

*Interpretability*. We consider coarse-graining over tokens, weights, activations, or the data space equally within scope. Finally, in Section 2, we propose research objectives to guide interdisciplinary work going forward.

The details of the renormalisation framework we depend on throughout this draft will be sparsely cited, and can be found in any introductory textbook on the topic (e.g., Peskin & Schroeder (1995); Amit & MartíN-Mayor (2005); Nelson & Zhang (2025)). We aim to be as heuristic as possible while crafting our position in a NN setting, without sticking to any particular paper or theoretical premise, and point the reader to Appendix D for supporting work.

## 2 BACKGROUND

### 2.1 RENORMALISATION: A PRIMER

In a physics context, renormalisation plays two main roles: to describe physical systems by *effective* theories at different scales of observation, and to decouple their degrees of freedom into a hierarchy of approximately local interactions by systematically identifying the *relevant* parameters as the scale is varied, so that fine-grained details are discarded while coarse-grained behavior is preserved. Starting with a theoretical description of a statistical system – for example, a Hamiltonian, or action specifying interacting degrees of freedom (e.g., spins, fields, or NN components[2]) and *couplings* that determine their interaction strengths – a renormalisation step can be thought of as an operation of two parts:

1. Coarse-graining: We average over, or 'integrate out', high-resolution degrees of freedom up to a cutoff scale[3] to obtain an effective description in terms of coarser variables. This requires choosing a sensible direction (like momentum or distance) along which to coarse-grain.

2. Rescaling: Re-express the fields and couplings so that the effective theoretical description – and the observables, or measurable quantities, it predicts – remain valid at the new, macroscopic *scale of observation*[4].

This is a factorization of a single coarse-graining operator that acts on the space of probability distributions or parameterized models, mapping them to progressively coarser models along some scale. Iterating this procedure results in a renormalisation Group (RG) flow: the couplings evolve between the microscopic and macroscopic scales, leading to a chain of effective field theories[5]. The map between these theories is generally nonlinear; integrating out high-resolution modes generates effective couplings that are complicated functions of the original ones. The flow can be summarized by a $\beta$-function: $\beta(g) = \partial g/\partial \ln(\mu)$, where $g$ is a coupling and $\mu$ is a scale. Linearizing this around a point along the flow gives a Jacobian in the space of couplings, with eigendirections that are relevant (grow under coarse-graining), irrelevant (shrink under coarse-graining), or marginal (remain stable). Typically, this classification is done close to a fixed point of the flow, where the couplings stop evolving with scale, but the same picture holds locally around any effective theory. We guide the interested reader to Appendix C for more background from physics and Appendix E for a glossary of key terms.

### 2.2 MULTI-SCALE STRUCTURE IN AI SYSTEMS

It is often said that NNs are grown rather than built; they exhibit behaviors more similar to systems studied by statistical physical rather than those that are human-engineered Allen-Zhu & Li (2025); Yang (2021); Ringel et al. (2025a); Bahri et al. (2024). This, in turn, suggests that they may be amenable to renormalisation techniques. For NNs, we find it useful to distinguish between *implicit* or *explicit* renormalisation. In Appendix A, we provide heuristic examples of both types, as well

---

[2]Because some of the literature refers to quantum or statistical field theory, we may refer to these components as 'field-like', in spite of differences between discrete NN parameters and continuous fields.

[3]This is often referred to as the UV, or ultraviolet, cutoff, as this corresponds with a high-energy limit.

[4]This is known as the IR, or infrared.

[5]We will use 'renormalisation' and 'RG' interchangeably in a NN context. This is a useful convention, and not a statement about (semi-)group-like structure within NNs.

as an evaluation protocol and potential failure modes. We stress that thinking of neural networks as coarse-graining information in various ways (i.e., across layers) is not new; our goal is to weave together various threads to serve our AI safety agenda. In Appendix D, we give a partial review of existing literature. These are mainly examples of implicit renormalisation, though our position is that there is high potential for application to the explicit case. We organize work according to how effective degrees of freedom (features) are defined: i) as kernel components and ii) directly in the dataspace. While many of the cases we consider apply in an idealized or toy model of data or inference, we strive for the more general application of these ideas. We note that many of the works considered use different terminology, and aim to be explicit about this.

***Implicit renormalisation***. During training and inference, NNs coarse-grain information about the data into 'model-natural' structures that reflect the inductive biases informed by a network's architecture, data distribution, and training details. Implicit renormalisations schemes describe this process, with scale and coarse graining arising from the model's own training dynamics. We do not expect there to be a single, canonical implicit scheme; different theoretical descriptions can track different scales and coarse-grainings (e.g., across depth, noise level, or context length, during training or inference). Examples include diffusion models and language models.

***Explicit renormalisation.***The goal of interpretability is to design post-hoc tools that are both faithful to the model ('model-natural') and human-interpretable. With explicit renormalisation, this goal is achieved by relating faithfulness with an implicit renormalisation hypothesis. Resulting tools use explicit scale parameters and coarse-graining rules to make sense of model internals (e.g., weights or activations), leading to a multi-scale model of *interpretations* that reflects their learned, multi-scale structure.

In section B.4, we propose a pair of research artifacts aligned with these two flavors of interpretability.

## 3 RENORMALISATION FOR INTERPRETABILITY

For each RG-like scheme developed in an NN context, there are three important questions to address:

1. *How* we coarse grain. What defines an RG-like coarse-graining scheme for NNs? Which model-natural notion of scale does it track, what is the metric for relevance, and in which empirical settings is that scheme robust?

2. *What* are the inputs to, and properties of, an effective description that comes out of a particular coarse-graining scheme? Given a choice of cutoff, what are the relevant features for a given computation, and which can really be treated as irrelevant-small fluctuations? How far is an effective description from a critical point?

3. *Why* does this matter for interpretability? What experimental signals correspond with safety-relevant observables? Under what conditions does separation of scales hold? To what extent do universality classes meaningfully constrain or describe a model's behavior?

We develop these in more detail in Appendix C. In this section, we lay the groundwork for the last point. We imagine modeling NNs and the data they represent via a hierarchical decomposition of components (e.g., features) that depends on a coarse-graining resolution, or scale. This is an intentionally broad operationalization, meant to keep our focus on the *interpretability goal* rather than any particular renormalisation scheme. Simply put, we aim to produce interpretability tools with renormalisation-theoretic guarantees, which cleanly separate relevant from irrelevant variables in a context of interest, with bounded error. This is the 'separation of scales' property mentioned earlier: microscopic details (e.g., individual parameters or finer features) can vary within some range without materially changing the coarse behavior of a chosen macroscopic description (e.g., a downstream behavior or set of aggregate features).

Worries about bleed-in from other scales and contexts are frequently observed in the AI safety literature. These include i) the use of steganography in chain-of-thought, where information is hidden in apparently random tokens(Karpov et al., 2025), ii) Bayesian assumptions about independence or Gaussian noise terms that collapse under distributional shift (Christiano et al., 2022), and iii) Causal feature graphs that track statistically relevant circuits in some contexts but are sensitive to slight changes in the input distribution (Marks et al., 2025).

In addressing these, we are not merely suggesting a reframing of existing desiderata. Current tools optimize for reconstruction accuracy or human interpretability, but are incapable of making rigorous claims about causal necessity or insufficiency, let alone make guarantees about what they miss. A useful separation of scales argument would take the form: 'conditional on an effective description, (potentially catastrophic, low probability) perturbations confined to the irrelevant subspace cannot change observable $X$ more than $\epsilon$. It is our view that a lot of the work in Appendix D is closer to being turned into a theory-driven, SOTA-scale interpretability tool than one might expect.

We stress that this separation of scales property is non-trivial. While we may formally introduce any coarse-graining scheme on any statistical system, only some can be compressed in a way that shields – in a way that holds across scales – the long-range effect of a small set of relevant parameters from short-range details. In physics, such systems are said to be renormalizable. A proof of separation of scales is possible in many high energy or idealized condensed matter theories. In other systems, it can nevertheless be shown to hold empirically (for example, by examining scaling laws)(Cassandro & Olivieri, 1981). Which of RG's many theories and techniques – including separation of scales – can be imported directly, and which should be adapted for NNs, is still an open question (see D).

## 4 DISCUSSION

This paper argues that renormalisation offers a productive lens for ambitious interpretability, and that the field has matured to the point where scattered research threads can be synthesized into practical tools capable of making worst-case guarantees. Several developments since we began this work suggest growing momentum in this direction as a result of intentional coordination efforts. For kernel renormalisation, these include less idealized pictures of renormalisation and universality in NNs (Coppola et al., 2026) and a hypothesis for feature identification using the eNTK (Lin, 2025). For data-space renormalisation, Brill (2026) presents a code repository for generating synthetic datasets that encode hierarchical data structure and Berman & Stapleton (2026) presents a candidate model-natural scale based on tokenization. There has also been work, to appear soon, developing probabilistic SAEs that leverage the hierarchical structure of DAGs (Mack et al., forthcoming).

We are not alone in thinking that building better tools goes hand-in-hand with better theory development, or that insights from physics can make progress in AI safety. We hope that the work discussed here will provide productive points of contact with related efforts to close the theory-practice gap within the field. These include the mechanistic modeling of belief states in a transformer residual stream, using insight from computational neuroscience and theoretical machinery from computational mechanics (Shai et al., 2025), the study of how learning dynamics and generalization depend on data, inspired by singular learning theory (Adam et al., 2025), and work aiming to formally explain neural network behavior to detect potentially harmful anomalous or low-probability behaviors, rather than focus on the average-case scenario (Wu & Hilton, 2025b).

***A Call to Action***
As a framework, renormalisation is a cornerstone in the explanatory powerhouse of modern theoretical physics, capable of capturing the essential empirical aspects of a system and how its theoretical description fits in with the space of possible theories. The work surveyed in Appendix D spans computer science, physics, biology, and complex systems science – fields with different vocabularies, tools, and publication venues – and remains under-formalised in an AI safety context. We think this is a missed opportunity to put one of physics' most flexible foundational frameworks to work on one of today's most pressing problems.

We do not advocate for importing physics wholesale, but for the careful translation of concepts – scale, relevance, and separation of scales – so that they can make robust guarantees for real-world AI systems. Concretely, in B.4, we propose a path to impact centered around a pair of research artifacts around which researchers from across disciplines can focus their efforts. We believe a deliberate effort to bridge these communities, by sketching the problem (from AI safety) and potential solutions (from renormalisation) in a more neutral language, will accelerate progress so that it can keep pace with threats from AI systems.

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

## A  RENORMALISATION HEURISTICS

### A.1  IMPLICIT RENORMALISATION

We consider ***diffusion models***(Ho et al., 2020) to be a prime example of implicit renormalisation. Here, what we call 'scale' would be the diffusion time or noise level. As for coarse-graining, the forward noising process gradually adds noise to clean data, washing out fine details to model stable, coarse patterns (e.g., global composition, abstract shapes). A denoising step, implemented by a neural network, reverses this process, progressively resolving finer-scale features (e.g., textures, edges) by predicting an image at a slightly less noisy scale. Here, the architecture and training objective explicitly impose a scale hierarchy as an ordered sequence of noise levels, but the way information is represented and reused at each scale is learned implicitly during training. This suggests that relevant features of the data are those that remain predictive across many noise levels. Diffusion is one of the clearest empirical examples of renormalisation-like behavior in AI systems, but that does not mean the parallels with statistical physics are complete or exact. Instead of a flow of an equilibrium ensemble as its scale of interaction changes, diffusion is a trained, approximately invertible noising–denoising process on samples. We treat it as RG-like because noise level defines a scale hierarchy and organizes coarser versus finer descriptions of the data distribution.

In a ***language model*** setting, there are many ways we think about coarse-graining. *Candidate Scales* include data granularity (e.g., context length) or eigenmodes of latent features (e.g., correlations of activations or the NTK, see Section D.1.1). Language models implicitly organize information across scales, tracking slowly varying context variables (setting, speaker identity, topic, emotional tone) that remain stable across long stretches of text, alongside finer-grained details like grammar, syntax, and token-level semantics. That they learn such multi-scale features suggests that internal activations and attention patterns are sensitive to a text scale, and that a latent hierarchy of features

emerges during training and adapts during inference. Unlike diffusion, language models are not explicitly trained to coarse-grain over the data scale; the latent hierarchy need not coincide with, for example, spectral scales defined by kernel eigenmodes. Heuristically, this points to multiple model-natural scales that could organize features in terms of their relevance to a given task. Relating or combining these is a core problem for the type of scale-aware interpretability we have in mind. Sun & Haghighat (2025) gives an analogy between transformers and the 1-dimensional Potts model, and Peebles & Xie (2023) introduces the diffusion transformer, a robust architecture that puts phenomena related to renormalisation and lattice models in the forefront.

## A.2 Explicit renormalisation

There is much less relevant literature for explicit renormalisation, so we paint a picture here of what we have in mind. In an ***Interpretability tool***, for example, increasing the hidden dimension in SAEs often yields finer-grained semantic features—a coarse 'scientist' feature splits into Einstein, Turing, Goeppert-Mayer. This resembles RG flows where previously degenerate observables separate at finer resolution, suggesting that such tools are capable of probing data-model interactions at a tunable scale. This scale is likely more complicated than a single hyperparameter; understanding when such explicit decompositions respect, distort, or completely miss the model's implicit scales is one of the problems that a renormalisation-guided perspective is meant to clarify. Matrioshka SAEs Bussmann et al. (2025) were designed partially to combat this problem.

Other heuristics for explicit renormalisation methods include: clustering or pruning neurons, truncating spectral decompositions (e.g.., keeping only the top-k kernel eigenmodes), or applying information-theoretic compression schemes. Each of these methods defines its own notion of scale (e.g., number of clusters or sparsity, eigenvalue cutoff) and relevance (which features are treated as important for explaining behavior).

## A.3 Evaluation Protocol and Limitations

We acknowledge significant uncertainty about which RG-inspired methods will transfer. First, identifying model-natural scales is largely empirical. Second, renormalisation workhorses like criticality may not hold in NNs, or may hold only in restricted regimes. It is also possible that the computational cost of GRT may limit their scalability to SOTA models, and worst-case guarantees that hold in idealized cases may weaken in messier, real-world settings. We expect that similarities between physical and AI systems will lead to a rich analogy rather than a strict correspondence, underscoring the need for care in selecting research directions with the potential for impact in AI safety and effectively communicating this safety relevance, as well as any physics jargon.

In evaluating an RG scheme, we consider the following failure modes. If discarding small-scale components significantly degrades performance or otherwise impacts long-range observables, it may indicate (see section 3 for more details):

- The scheme assumes a wrong scaling direction that does not reflect the model-natural hierarchy.
- The chosen cutoff is not tuned to the resolution at which model components (e.g., effective features) are observed.
- Our notion of relevance is misaligned with the level of detail a computation needs.
- We've encountered a dangerously irrelevant feature that shrinks under coarse-graining but whose fluctuations has a definitive impact on large-scale behavior[6](Cardy, 1996). This highlights the importance of choosing observables that properly constrain the feature class.

A scheme that fails along one of these axes signals a mismatch between our assumptions and the model's true structure. Identifying which failure mode applies in an empirical setting can guide theoretical refinements to the coarse-graining procedure or, potentially, reveal regimes where worst-case guarantees are inherently out of reach. We think that even qualified success of the proposed agenda has the potential to build a varied, adaptive framework for AI safety.

---

[6]These may be irrelevant with respect to the fixed point of interest, but relevant with respect to nearby fixed points. Setting these operators to zero can therefore alter the overall structural properties of the RG flow. They also have implications on universality arguments and the calculations of critical exponents.

## B  ROUTE TO IMPACT: A CASE STUDY

We are inspired by the recent history of how an exploration of the superposition hypothesis – the idea that models represent more features than directions in activation space – led to the development of sparse autoencoders as a tool to find interpretable features in transformer-based models. We present this story as evidence for the potential route to impact of a novel theoretical perspective. In this case, AI safety researchers imported Compressed Sensing (Donoho, 2006), an idea from applied math, along with its workhorse, dictionary learning (Tillmann, 2015), to make the story come together. It offers a compelling case study in mechanistic interpretability in which empirical observations and well-posed ideas led to a measurable phase change in how we frame, analyze, and interpret NNs. Its progression is marked by two key outputs.

### B.1  OUTPUT 1: MODEL ORGANISM OF SUPERPOSITION

Neural networks are polysemantic – their constituents (neurons, attention heads) are observed to activate in multiple distinct contexts. In Toy Models of Superposition (TMS) (Elhage et al., 2022), their seminal work on the topic, Anthropic explores superposition as a hypothesis for this phenomenon by studying how models trained on data with ground-truth features can represent more of those features when they interfere. Central to TMS are two ideas:

1. *Sparsity.* Each input is a linear combination of features, defined here as vectors in the input space. Superposition occurs when there are more features with significant statistical weight than there are neurons. TMS encourages this by tuning the sparsity in the input space; when sparsity is high, each feature appears in fewer training examples. To minimize the loss, the model compresses multiple features into overlapping directions in activation space.

2. TMS also relies on the assumption that features are linearly represented, known as the linear representation hypothesis (LRH) (Nanda et al., 2023b; Mikolov et al., 2013). This essentially defines a 'feature' as a direction in activation space, one that ideally preserves the structure of the input space – this is how they defined a feature in the toy setting. Though directionally useful, this approximation may break down in more realistic settings, making the LRH is at best a useful approximation (Engels et al., 2025b).

TMS is essentially a model organism for superposition – a toy neural network setting designed to encourage superposition implicitly in its internal representation. This simple picture allowed researchers to refine and probe the superposition hypothesis and what it means for NN training, inference, and feature geometry. Compressed sensing offers useful guides in their analysis, including bounds on the number of features in superposition (Donoho & Tanner, 2009).

### B.2  OUTPUT 2: SPARSE AUTOENCODERS (SAEs)

Underlying the superposition hypothesis and its relation to polysemanticity is that there exists a collection of linear directions in activation space (features) which are monosemantic, i.e., correspond to single atomic concepts. If this is true, and if those directions could be found, it would be a boon for interpretability. Enter the SAE: a tool built on dictionary learning – a compressed sensing technique – which learns an overcomplete feature basis by sparsely reconstructing activation data (Bricken et al., 2023; Shu et al., 2025).

The addition of a sparsity constraint to the loss function tweaks the LRH to a new hypothesis which supposes that there exist linear, monosemantic feature directions that explain the activation data, and that these features are activated sparsely by each activation. Validating this hypothesis involved generating two kinds of (empirical) evidence:

1. *Statistical competitiveness:* Earlier unsupervised attempts to rotate the neuron basis, like PCA, revealed some important directions but failed to produce monosemantic units except for some cherrypicked cases. SAEs were shown to outperform these baselines (notably, in transformers). More precisely, activations could be compressed more efficiently by assuming sparsity in the SAE basis than in PCA or randomized control bases with matched second moments. While this does not guarantee interpretability, it provides strong evidence

that the hypothesis is a good statistical fit. We refer the reader to Karvonen et al. (2025) for a review of SAE benchmarks, a discussion of their limitations.

2. *Semantic Interpretability:* Even if sparse decompositions exist, they are not guaranteed to be monosemantic in a way that is human-interpretable. But SAE features turned out to be remarkably so, at least compared to any other unsupervised way to get linear features (Cunningham et al., 2023). This was checked in a number of ways, including using probes (Gurnee et al., 2023; Alain & Bengio, 2018) to select for dataset examples that strongly activate a feature and activation addition (Turner et al., 2024) (e.g., adding a feature to a hidden layer and observing how it affects the model output). These methods revealed that many learned directions correspond to apparently monosemantic concepts, as measured by an automated (LLM-evaluated) interp score and by eye (Paulo et al., 2025; Lin, 2023).

### B.3 The Takeaway

While the progression from TMS to SAEs was productive for AI safety, it points to several limitations that motivate the need for better hypotheses, tools, and techniques:

1. *The LRH is approximate.* TMS assumes that features correspond to directions in activation space, which reflect linear features in the data. While this provides a tractable mathematical setting, it likely misses contextual, hierarchical, or nonlinear structure present in real data (for example, modular addition (Nanda et al., 2023a)). SAEs build on this view by using linear dictionaries trained under sparsity constraints to reconstruct NN activations, which can be informative but are not guaranteed to reflect the model's own (potentially nonlinear) internal structure (Engels et al., 2025a; Csordás et al., 2024; Hindupur et al., 2025).

2. *"Feature" is ambiguous.* The notion of a "feature" is underdefined and used inconsistently in AI safety.

   • In TMS, features are (in this case, ground-truth) directions in the input space. The model learns to represent these in activation space, but whether these internal directions match the true data features depends on training regime and inductive bias—it's not guaranteed.

   • In SAEs, features are dictionary directions in activation space that reconstruct model activations under a sparsity constraint. These are assumed to correspond to atomic, monosemantic units, but this is typically justified in an ad hoc or unprincipled way. This definition is further muddied by phenomena (like feature splitting) discussed in the 'limitations of current SAE methods' below.

3. *'Interpretable' is ambiguous.* Monosemanticity is often taken as a stand-in for interpretability, but are these definitions synchronous? While TMS observes a sharp phase transition between monosemantic and superposed regimes, a monosemantic feature might correspond to a fuzzy, uninterpretable internal concept even if it encodes a statistically pure direction. Conversely, an interpretable feature might be polysemantic in structure but contextually well-understood.

4. *Current SAE Methods are limited.* While SAE-inspired approaches are currently the most scalable and principled tools for feature decomposition, they fall short of the criteria we consider essential for a robust interpretability toolkit:

   • **Canonicity**, incorporating:
     – *Uniqueness:* Different SAEs (datasets, hyperparameters, training runs) can produce different decompositions(e.g., Paulo & Belrose (2025)).
     – *Completeness:* They may omit significant but entangled features (e.g., (Hindupur et al., 2025)).
     – *Atomicity:* The features are not guaranteed to be irreducible (feature splitting, absorption (Chanin et al., 2025)). Theoretically, SAEs are too weak to capture certain important atomic behaviors(e.g., (Leask et al., 2025)).
   • **Faithfulness**: There's no guarantee that the features reflect the model's causal structure or computations. Causal scrubbing and other techniques for validating decompositions reward loss-preserving approximations, not faithful reconstructions of the model's actual internal mechanisms (Chan, 2022; Geiger et al., 2025). This raises the

risk that tools like SAEs may capture statistically predictive proxies rather than the true causal structure of model computation or reasoning.

Additionally, SAEs optimize for a trade-off between reconstruction accuracy and compactness (description length) of an explanation. The extent to which a sparsity prior selects for efficiency over canonicity or faithfulness is a core question in interpretability work (Karvonen et al., 2025).

In spite of its limitations, we think that this story exposes a model for research that, if duplicated, could narrow the theory-practice gap and drive progress in ambitious interpretability. Key components of this model include:

- *A tight theory–experiment loop:* The superposition hypothesis led to testable predictions in TMS, which then catalyzed the SAE method for probing real models—a quick and tightly connected pathway from concept to tool.

- *Cross-Disciplinary Agility:* TMS successfully adapted techniques from compressed sensing to the problem of feature decomposition, sharpening both the theoretical framing and the empirical setup, and exemplifying how ideas from applied math and signal processing can be translated into AI safety contexts. Knowing which aspects of an external field are relevant, which are not, and how best to translate them to a new application, is a core but worthwhile challenge in cross-disciplinary collaborations.

- *Rapid iteration:* The time from the initial TMS paper to the adoption of SAEs as a tool in mainstream interpretability research was under a year. This reflects a nimble, experimental style of science: start with a tractable toy model, iterate quickly, and scale up promising insights.

- *Clarity of purpose:* The work was animated by clear goals: understanding polysemanticity via superposition and finding interpretable monosemantic models. Each step in the pipeline was framed by these guiding questions.

- *Aiming for tools that scale:* SAEs are computationally tractable and more or less efficient, relatively easy to apply and scale across layers and models. The value of toy models disappears if insights fail to transfer in realistic settings.

Using this model, our goal is to develop new, testable hypotheses that lead to interpretability tools without the failure modes described in the last section. Statistical physics, and renormalisation in particular, could provide a theoretical and empirical framework for *defining theory* and *employing tools* that leverage meaningful structure that current tools lack. For example, we aim to work with a definition of 'feature' that is not predicated on the LRH, capable of handling non-linear, compositional, and hierarchical structure. This should be both model-natural and interpretable, able to capture partial, contextual, or compositional structure. In classifying or interpreting features, we aim to prioritize **relevance** over canonicity, seeking effective descriptions that summarize behavior at a given scale of abstraction. We hope that additional properties that a renormalisation framework adds – like separation of scales – will improve the **faithfulness** of a given set of features.

## B.4 A Call to Action

Inspired by this story, we propose a pair of research artifacts to guide interdisciplinary work going forward. We sketch these below, and leave their development for future work.

***Artifact 1: Toy Model of renormalisation (TMR).*** We aim to develop a model organism of renormalisation analogous to what TMS was for sparsity. Rather than a model for how features interfere, a TMR should generate a hypothesis for how they compose, coarsen, and depend on scale. This artifact maps onto implicit renormalisation, which focuses on the development of robust theoretical descriptions for empirical phenomena. A moonshot result would develop a renormalisation-inspired hypothesis capable of describing training and inference in a way that provably bounds the influence of fine-grained components on safety-relevant behaviors. Along the way, we will piece together rigorous clues to explain phenomena in individual settings (e.g., across tokens, in information space, in various kernels, or in the loss landscape) through the lens of scale separation, operator relevance, or effective degrees of freedom. It is likely that this will involve designing synthetic data distributions with ground truth hierarchical structure and studying the learned representations of models trained

on them; such settings can also act as empirical validation methods for desirable properties in interpretability tools. We advocate for scalable insights that are well-contextualized in terms of their assumptions and regimes of validity.

***Artifact 2: General renormalisation Tool (GRT).*** Mapping onto explicit renormalisation, our second goal is to build a general-purpose tool – analogous to SAEs – that extracts interpretable, multiscale structure from real models. Following the logic of Appendix B, GRT should reflect how TMR models the data, weights, or other associated statistical artifact. For example, it could apply lattice RG to activation graphs, construct coarse-grained feature maps from causal states, or a flow defined by Polchinski-style equations(Polchinski, 1984). The computational analogy is not "find atomic parts and interpret them" but "find effective descriptions that represent meaningful, interpretable structures as a function of a model-natural scale." In this case, the holy grail, aligned with a sweeping renormalisation hypothesis, would be a completely novel tool capable of outperforming SAEs on all desiderata like canonicity and faithfulness. Instead of coming up with this out of the gate, we imagine an iterative process where theory and practice approach each other incrementally, starting by formalizing renormalisation-like behavior in existing tools.

## C BACKGROUND AND DESIDERATA

### C.1 HOW? DEFINING RG-LIKE COARSE-GRAININGS IN NEURAL NETWORKS

In physics, there is a clear notion of scale (distance, energy, or momentum) attached to a physical hierarchy of particle interactions. This guides the development of an RG scheme: spins on a spatial lattice are grouped into blocks in real space while momentum shell methods like Wilsonian RG or Polchinski RG are more natural for a particle system in space-time. Each scheme depends on a cutoff scale which determines how much to coarse grain at each point along an RG flow. In lattice models, there is both a maximum distance (the system size) and a minimum distance (the lattice spacing), which effectively grows with the number of coarse-graining steps. Other RG schemes systematically remove high-momentum modes for particle hierarchies organized according to this scale. In either setting, fine-grained degrees of freedom beyond the cutoff scale are ignored because they don't contribute to phenomena of interest, marking a physical limit of the effective theoretical description[7].

Keeping with the distinction we made earlier, we define two types of cutoffs for NN renormalisation. Implicit cutoffs arise during training and reflect finite resources of the model. Given a particular dataset, architecture, and optimization procedure, the trained network only represents a restricted subset of possible functions, resolving structure in the data up to some effective resolution. We conjecture that this structure reflects an implicit feature hierarchy along some model-natural notion of scale[8]. There are empirical indications of this, including spectral gaps in kernel eigenvalues and low-variance directions in activation space. While these patterns likely reflect partial aspects of a true data feature hierarchy, they indicate that, for a given task, many directions are treated by the model as irrelevant small fluctuations (see Section D.1).

In contrast, explicit cutoffs are imposed on the system by interpretability or analysis tools. PCA, SAEs, clustering methods, or graph-based coarse-graining procedures all introduce additional scales, defining relevance according to, for example, sparsity, dictionary size, variance explained thresholds, or graph resolution. These choices determine the finest scale at which we attempt to resolve features in an explicit decomposition. Beyond that scale, additional components may be too fine-grained for the description we want (as in the Ising model), unstable under small changes, or not meaningfully interpretable (as in the standard model). In a renormalisation-guided framework, a good explicit cutoff should respect the implicit cutoff of the trained model; the resolution of our interpretation should line up with the finest scales at which the network has actually learned structured features, rather than artificially abstract finer details (though it may cut off fine details for a coarser description). We aim for future renormalisation-based tools to provide a practical and safety relevant cutoff on interpretability resolution (see Section D.2).

---

[7]A cutoff can also mark an epistemic limit beyond which a new, more complete theory is needed to describe physics beyond that scale. The standard model, for example, is widely regarded as an effective truncation of a theory of quantum gravity.

[8]Some candidate examples of scales in specific settings were given in section 2.2.

In addition to a cutoff, any RG-like construction rests on an implicit notion of locality, which determines the range of interactions and gives a measure for closeness between degrees of freedom. In physics, most interactions are short-ranged, or local, in space-time, though non-local interactions do arise, particularly in effective field theories. Importantly, renormalisation is constrained such that locality is approximately preserved[9]. In NNs, 'nearby' depends on context; in input space it could be neighboring pixels or tokens, in data space it could be according to some graph structure of natural language (see Section D.2), and in representation space it could be a kernel-induced distance (see Section D.1). A good explicit renormalisation scheme should probe and preserve an implicit notion of locality (see Sections D.2.1 and D.2.2 for disucssions of locality in the input space and data space, respectively). If our understanding of scale and locality is badly misaligned with how the model actually couples its degrees of freedom, coarse-graining could produce cryptic or highly nonlocal effective descriptions, undermining the interpretability guarantees we seek.

Finally, an RG-like scheme requires a direction along which degrees of freedom can be collectively summarized. These are not as obvious in NNs as in nature[10], but just as power laws of operators do in physics, these can be guided by candidate measures of local interactions that exhibit power-law behavior (see Section D.2.2). We stress that individual hyperparameters (e.g., width, learning rate, dataset size, initialization variance, SAE dimension) are not what we mean here, though these help shape the inductive biases that give rise to a model's implicit hierarchy. These quantities are better thought of as control parameters that select broad regimes or basins of attraction in model space (e.g., NTK-like v. feature learning regimes, see Sections D.1.1, D.1.2, and D.1.3), rather than directions for coarse-graining. We see model-natural scales as something to be discovered and justified as the renormalisation framework is developed. A single, model-natural coarse-graining scale, if it exists, is likely to be a combination of the various scales we can currently track, such as ordering kernel eigenmodes by eigenvalue, evolving diffusion time or noise level, tracking text granularity, or organizing representations across layers by some summary statistic (see Sections D.2.4 for an information-theoretic view of scale, D.2.5 for a discussion about scale and relevance framed in terms of statistical inference, and D.2.6 for a discussion on compressibility and interpretability). Different choices will typically induce different flows, though some may converge to similar effective descriptions. In physical systems, many different notions of scale and RG flow lead to the same renormalized theory; in some sense, the space of RG flows with the same limit is open in a suitable topology, and quite forgiving in practice[11].

In practice, there are two common viewpoints on RG from physics: Wilsonian RG, which operates in momentum-space, acting on functions and tracking couplings as they flow along a continuous parameter, and real-space RG which is often applied to discrete systems and explicitly removes degrees of freedom in a stepwise fashion (for example, via Kadanoff decimation on a lattice). Though a lot of existing work takes a Wilsonian view, we will not discount real-space methods, which may still prove useful for coarse-graining in certain ML settings (see, for example, Section D.2.4).

C.2   WHAT? EFFECTIVE DEGREES OF FREEDOM AND RELEVANCE

Once specified, an RG scheme is a transformation from one effective description to another – it takes in a messy, high-dimensional model and reduces it to a smaller set of effective features whose behavior is enough to predict the phenomena we care about, up to a specified error.

A useful way of thinking about this is through spectra. In a standard Wilsonian picture, as we change scale, the spectrum of operators reorganizes so that a small set of low-dimension operators carries most of the weight for long-distance observables, while the rest are effectively decoupled. In an analogous picture for neural networks, a handful of modes in a kernel spectrum acquire large

---

[9]Coarse-graining may generate longer range interactions, but under a well-defined RG scheme these are suppressed at large scales.

[10]To first order, the local approximation of the $\beta$-function yields a linear rescaling of the units used to make measurements, ensuring dimensional consistency. In NNs, where everything is dimensionless, analogous rescalings must be based on other – currently unknown – principles.

[11]This is particularly visible in the critical Ising model and its Kramers–Wannier dual: defining a Gaussian convolution RG in the fermionic representation of the critical Ising model yields a natural, local smoothing procedure. However, mapping this RG explicitly back to spin variables produces a well-defined but inherently nonlocal RG scheme. Thus, while the duality preserves the universality and consistency of this RG flow, its interpretation and locality properties differ dramatically between the dual descriptions.

eigenvalues as we change scale or training regime, while the remaining modes collapse into a low-variance tail. For a given observable and error budget, only the top modes are genuinely relevant, and the tail can be safely neglected.

More generally, a renormalisation-like coarse-graining scheme should take in:

- *A set of features.* In interpretability research, the term 'feature' is used in two related ways: as a ground-truth property of a data distribution that a model can learn, and of the model component (e.g., activation) that represents that property. Similarly, we define this term loosely to mean any model-natural effective component that captures meaningful structure in data and representations, i.e., a derived variable computed from model internals (model inputs, weights, activations, or gradients) that can be tracked under coarse-graining and related to observables. These can be functions of a single-input (e.g., activations, circuits) or a pair of inputs (i.e., kernel eigenmodes).

- Depending on the setting – whether we are considering renormalisation in data space, input space, or activation space – and type of RG scheme (implicit or explicit), features can be built from inputs, weights, activations, or gradients. Examples include directions in activation space, eigenmodes of a kernel or covariance operator, and components of a hierarchical data model (see Appendix D for examples).

- *A model-natural notion of scale and cutoff that separates 'fine' from 'coarse' features.* In an implicit RG scheme, the cutoff is an input to the theoretical description that can be tuned to match the empirical structure that training and inference dynamics have imprinted on representations and kernels (e.g., an observed eigenvalue cliff). In explicit RG, it is an input to a post-hoc tool used to compress that structure (e.g., a constraint on the number of features per neuron).

- *One or more long-range observables and an associated error budget.* These are measurable quantities or quantifiable behaviors that should remain approximately invariant under coarse-graining, up to an acceptable tolerance for error. These could be related to performance on a specific task (e.g., next token prediction), probes of safety relevant behavior, or structural properties of internal activations (e.g., response functions (Baker et al., 2025)) that we wish to preserve. As in physics, we expect many observables to be global or structural quantities guided by empirical scaling laws (Brill, 2024).

Given these, the map should return:

- *A reduced set of effective features.* These constitute a valid description up to the cutoff. While this should be smaller than the original set of features, it may still be high-dimensional.

- *A relevance ordering on those features*, determined by their contribution to large-scale observables, depending on the cutoff. For implicit RG, this could result in some spectral separation for a downstream task. In explicit RG, this is heavily reliant on the architecture and optimization metric of the tool at hand, but should also reflect the implicit structure.

- *Bounds on the influence of neglected components.* For worst-case guarantees, a minimally rigorous argument would be of the form 'conditional on an effective coarse description, components above the cutoff have at most small, quantitatively bounded influence on the observables'. [12]

Crucially, relevance is not an absolute notion. It is defined relative to the scale at which we observe or intervene on the system, as this is where we measure observables. In physics, these are often defined by special points[13] in the – sometimes high-dimensional – space of couplings that control the dynamics of nearby effective theories. Many microscopically distinct theories can flow toward the same macroscopic fixed points; they share the same long-distance observables, a property known

---

[12]This is a probabilistic statement about the joint distribution of degrees of freedom across scales that we think can be reasonably achieved. Ideally, we'd like a stronger separation of scales argument that strictly bounds the impact of all components above a relevance threshold. This has implications on the kinds of worst-case guarantees we can make, which will be discussed in the next section.

[13]These can be fixed points or critical points that are stable, unstable, or saddle-like of the RG flow. At a fixed point, the $\beta$-function vanishes, and continued coarse-graining leaves the effective theory unchanged.

as universality[14]. If we did not set a long-range scale at which to stop coarse-graining, continued iteration on NNs would lead to meaningless noise similar to a trivial fixed point. Instead, we tie the endpoint of the RG flow to the scale at which we measure things like task performance, elicited behaviors, or training dynamics.

We can view effective degrees of freedom as local coordinates adapted to the flow near a given scale. Around each point in theory space, some directions grow (are relevant) under coarse-graining, while others shrink (are irrelevant) or stay the same (are marginal). We expect a similar picture for NNs. A feature can be relevant at one scale and irrelevant at another, or relevant for one class of behaviors and not for another. During coarse-graining, the effective description can change qualitatively: different combinations of features become important for the observables we track, while others collapse into noise. We refer to these points where features become (ir)relevant as crossovers[15] (see Section D.1.4 for examples from the current literature involving phase transitions and scaling laws).

## C.3   Why? What We Want out of a renormalisation Framework for Interpretability

The power of renormalisation comes from its ability to link microscopic couplings, which depend on somewhat arbitrary choices of cutoff and renormalisation scheme, to macroscopic observables–quantities that can be measured and tracked across scales. A correct application of the renormalisation toolkit to AI interpretability could serve two goals: as a microscope with a dial to hierarchically decouple different scales, and as a diagnostic that reveals interesting large-scale phenomena such as phase transitions and the emergence of new structures (see, for example, Section D.1.5). This would let us faithfully summarize microscopic model details without having to track every parameter individually, allowing for both ambitious interpretability that doesn't depend on the definition of an arbitrary atomic unit.

### C.3.1   Separation of Scales and Hierarchical Emergent Structure

Not every coarse-graining scheme has the properties we want. Many formally valid coarsegraining flows produce effective theories that are non-local in undesirable ways or else lose information about the observables we care about[16]. As with any tool or method, conditions, caveats, and regimes of validity are all part of the evaluation process. For AI safety, a key property we'd like to impose on a good renormalisation scheme is separation of scales.

This property is crucial for the last two RG outputs we discussed in the last section: i) the identification of effective features that contribute most to a safety-relevant observable and ii) the empirical or theoretical bounds on how much neglected, fine-grained fluctuations can impact that observable. We refer to this last point as hierarchical conditional independence, a Markov-like property of an RG flow that preserves the hierarchy of effective theories along some scale. This means that the effective theories, and the interactions they describe, can really be treated as independent since coarse variables at a given scale act as sufficient summary statistics for finer variables at the next, more microscopic scale[17]. Physicists often impose this requirement by ensuring that RG schemes preserve locality across scales, which puts quantitative bounds on short-range fluctuations generating anomalous large-scale behavior.

Our goal is to make similarly rigorous statements for NNs. In this setting, only coarse-graining procedures that robustly preserve a sensible hierarchy of feature interactions, rather than scrambling them, will be capable of making useful guarantees for AI safety. Coarse variables might be effective features at some resolution (e.g., top directions of a kernel or Fisher information matrix, or top activating SAE feature). Conditioned on these, hierarchical conditional independence means that, for a given observable, fine-grained activity in a local region – such as low-variance directions or

---

[14]We will discuss the implications for universality of NNs in the next section.

[15]Though crossovers can be critical points marking phase transitions, this is not a requirement. They can simply be a point between two effective theories in the same phase, with different feature partitions.

[16]For example, a naive spin decimation RG scheme for the two-dimensional Ising model yields an exact but highly non-local effective theory with long-range correlations. One can truncate these by hand to recover locality, but then the coarse-graining no longer preserves observables exactly.

[17]This property is robustly held in momentum-space RG, but is perhaps easiest to see in real-space RG on a lattice, where block spins are literal aggregates of component spins whose effects are effectively screened off.

SAE features that have undergone a degree of feature splitting – can be safely discarded since they don't couple strongly across scales[18].

The effective decomposition of a complex system into independent hierarchical modules with this separation of scales property is similar to causal mechanistic decomposition, which factorizes a system's behavior in a way that preserves causal counterfactuals at multiple levels of abstraction (Geiger et al., 2025). A renormalisation framework built on this property – which would add a scale, a flow between abstractions, and quantitative bounds on when we expect it to hold – would represent a significant leap in provable interpretability research. Even an RG scheme for which scale separation is weakly or conjecturally satisfied could be extremely useful in bounding the extent to which potentially dangerous large-scale behaviors can hide in small-scale mechanistic interactions[19]. For example, if we can show that, for a particular model and observable, behavior is controlled by a tractable set of effective features, and that the influence of remaining features is bounded, we can show that any adversarial fluctuations known to exist in the irrelevant subspace cannot impact the observables by a known tolerance.

We expect coarse-graining schemes with a rigorous separation of scales property to hold in toy models and under restricted conditions, and aim to use these to build up our understanding toward worst-case guarantees that hold in realistic settings. Empirically, we can check that the bounded influence of perturbations confined to irrelevant directions (e.g., spectral tails) move our observables by at most some small tolerance. An evaluation criteria for this desiderata is faithfulness to these observables (and corresponding macroscopic description): does the scheme make correct predictions about how these transform under coarse-graining, and does it preserve them within the stated error budget? We may also ask if our RG scheme preserves locality, such that a coarse-graining does not generate uncontrolled long-range interactions. This is a more ambiguous desideratum for NNs than physics, but we can check whether coarse-grained features retain a semblance of a ground-truth feature hierarchy in toy settings[20].

### C.3.2 Universality: Bounding our expectations in NNs

Separation of scales is a core safety-relevant property for which we want to evaluate a potential renormalisation-like scheme. However, NNs are significantly different from physical systems, and we do not expect all aspects of a physics' renormalisation – like universality and criticality – to hold in a generic NN setting. For one, NNs have extremely large data and representation spaces; high-dimensional feature substances may be needed even at macroscopic scales. Nevertheless, any method to significantly reduce the number of dimensions in the data or representation would be practically valuable for interpretability, and it is worth thinking through what extra machinery a renormalisation framework could bring with it.

In physics, renormalisation is often tied to a discussion of universality: many short-range theories flow to the same fixed point under an RG flow. These are typically characterized by a small number of relevant parameters and exhibit the same large-scale behavior, forming families of microscopically different descriptions known as universality classes. Interpretations of NNs are similarly many-to-one, leading to questions about what universality could mean in this setting and how it relates to a renormalisation framework we build (see Section D.1.2). Can we make statements about whole classes of AI behavior based on a finite number of relevant directions? Under what conditions (if any) do these depend on a small number of features?

Any stable fixed point defines a universality class in the sense that many different microscopic models can share the long-distance behavior defined there. However, critical points—fixed points controlling continuous phase transitions—are special. There the correlation length becomes very large, and the scaling dimensions of relevant operators show up as critical exponents governing non-analytic, power-law behavior of observables over many scales. This makes universality more dramatic: very different materials or lattice models exhibit the same scaling laws near criticality.

---

[18]In other words, they are suppressed by some NN analog of locality and correlation length so that they only change coarse observables by a bounded, small amount.

[19]By 'dangerous', we mean the class of behaviors that significantly violate specified safety desiderata, such as deception. More broadly, we could potentially place bounds on any anomalous or low-probability behavior (Wu & Hilton, 2025a).

[20]We could also empirically track the correlation length.

Generic fixed points still organize phases and irrelevant operators, but the associated scaling structure is typically less striking, with finite correlation lengths and more analytic dependence on parameters. In addition to describing interesting physical phenomena, the theory at a critical point often becomes more tractable. In NNs, critical phenomena may manifest as sharp crossovers in which features dominate as the scale is varied, signaling qualitative changes in how the model represents information, similar to a phase transition (see Section D.1.4). Identifying such regimes can help identify regions where model behavior is particularly fragile, or where microscopic changes can have amplified effects on safety-relevant observables. Smooth crossovers – though harder to diagnose – may be equally important for understanding NN behavior.

If it is shown to hold, a universality-like property could guarantee that effective features and their relevance order should not change wildly under small changes in data, initialization, or architecture, for a fixed model family. These could be detected by asking: Do effective descriptions and relevance orderings change smoothly along a proposed scale direction, or do small perturbations lead to radical changes? However, there is no fundamental reason a NN theory space must have this fixed point structure, and it is unclear i) how close a given representation must be to a fixed point to place it within a certain universality class and ii) how stable these are to changes in hyperparameters (or continued training). If NNs do exhibit genuine universality in specific settings (e.g., the Gaussian Process of infinite-width limit, or specific scaling regimes), these could offer theoretically tractable regimes that can be interpolated or expanded into more realistic settings.

### C.3.3 UNIVERSALITY: BOUNDING OUR EXPECTATIONS IN NNs

Separation of scales is a core safety-relevant property for which we want to evaluate a potential renormalisation-like scheme. However, NNs are significantly different from physical systems, and we do not expect all aspects of a physics' renormalisation – like universality and criticality – to hold in a generic NN setting. For one, NNs have extremely large data and representation spaces; high-dimensional feature substances may be needed even at macroscopic scales. Nevertheless, any method to significantly reduce the number of dimensions in the data or representation would be practically valuable for interpretability, and it is worth thinking through what extra machinery a renormalisation framework could bring with it.

In physics, renormalisation is often tied to a discussion of universality: many short-range theories flow to the same fixed point under an RG flow. These are typically characterized by a small number of relevant parameters and exhibit the same large-scale behavior, forming families of microscopically different descriptions known as universality classes. Interpretations of NNs are similarly many-to-one, leading to questions about what universality could mean in this setting and how it relates to a renormalisation framework we build (see Section D.1.2). Can we make statements about whole classes of AI behavior based on a finite number of relevant directions? Under what conditions (if any) do these depend on a small number of features?

Any stable fixed point defines a universality class in the sense that many different microscopic models can share the long-distance behavior defined there. However, critical points—fixed points controlling continuous phase transitions—are special. There the correlation length becomes very large, and the scaling dimensions of relevant operators show up as critical exponents governing non-analytic, power-law behavior of observables over many scales. This makes universality more dramatic: very different materials or lattice models exhibit the same scaling laws near criticality. Generic fixed points still organize phases and irrelevant operators, but the associated scaling structure is typically less striking, with finite correlation lengths and more analytic dependence on parameters. In addition to describing interesting physical phenomena, the theory at a critical point often becomes more tractable. In NNs, critical phenomena may manifest as sharp crossovers in which features dominate as the scale is varied, signaling qualitative changes in how the model represents information, similar to a phase transition (see Section D.1.4). Identifying such regimes can help identify regions where model behavior is particularly fragile, or where microscopic changes can have amplified effects on safety-relevant observables. Smooth crossovers – though harder to diagnose – may be equally important for understanding NN behavior.

If it is shown to hold, a universality-like property could guarantee that effective features and their relevance order should not change wildly under small changes in data, initialization, or architecture, for a fixed model family. These could be detected by asking: Do effective descriptions and relevance orderings change smoothly along a proposed scale direction, or do small perturbations lead

to radical changes? However, there is no fundamental reason a NN theory space must have this fixed point structure, and it is unclear i) how close a given representation must be to a fixed point to place it within a certain universality class and ii) how stable these are to changes in hyperparameters (or continued training). If NNs do exhibit genuine universality in specific settings (e.g., the Gaussian Process of infinite-width limit, or specific scaling regimes, see Section D.1), these could offer theoretically tractable regimes that can be interpolated or expanded into more realistic settings.

# D A PARTIAL REVIEW OF EXISTING LITERATURE

Much of the existing literature provides examples of implicit renormalisation, though our position is that there is high potential for application to explicit renormalisation. We organize work according to how effective degrees of freedom (features) are defined: i) as kernel components and ii) directly in the dataspace. While many of the cases we consider apply in an idealized or toy model of data or inference, we strive for the more general application of these ideas. We note that many of the works considered here use different terminology, and aim to be explicit about this. This section surveys work examining renormalisation-like phenomena found in kernel structure and the data-space. We emphasize that this is a selective overview of directions we find promising, not a comprehensive review of the research landscape.

## D.1 KERNEL RENORMALISATION

We broadly view a kernel as a model-natural covariance operator on functions over inputs which organizes features as eigenfunctions (kernel modes). Kernels impart a notion of similarity to features in an input-dependent way, and offer a window into NN structure – the function spaces kernels have access to and the dynamics by which they evolve.

Two key kernels representing complementary perspectives dominate the literature: the Neural Network Gaussian Process (NNGP) kernel, which captures the prior distribution over functions at initialization, and the Neural Tangent Kernel (NTK), which captures how that function evolves during gradient descent. In certain limits (infinite width with appropriate scaling), these kernels fully determine network behavior. While these kernels have been studied extensively in this limit (also known as the infinite-width or 'lazy learning' limit), they have since been studied both empirically and theoretically in more general and expressive regimes [21].

Existing literature has grown rapidly, and differences in terminology and framing abound, with papers often using incompatible or context-specific notation and assumptions[22]. In an effort to standardize work going forward in a way that is useful for AI safety, we paint the following picture: kernel eigenfunctions correspond to features for the NNGP and feature tangent directions for the NTK. Coarse-graining then ranks features by relevance (to first order) according to their associated eigenvalues (prior variance for the NNGP and rate at which features are learned for the NTK). Choosing a UV cutoff is often like spectral truncation – considering modes below a scale that are relevant, for example, for a certain downstream task is like saying that nearby inputs (in the kernel-induced geometry) above this scale become indistinguishable at coarser resolution.

Even if it does not mention 'renormalisation' specifically, work from the physics community relating NN behavior with field theory has the potential to shed light on natural scales and notions of similarity and relevance in NN coarse-graining schemes. Future work should stress-test the limits of existing kernels and their underlying assumptions to understand when certain theoretical regimes (defined by initialization and hyperparameter choices) break down, and how to extend beyond them. Translating largely theoretical insights into practical interpretability tools would also benefit from greater understanding of the relationship between kernel features and SAE features. Could kernel features provide a renormalisation-based fix for SAE pathologies by capturing structure SAEs

---

[21]In the lazy learning regime, kernel regression with the frozen NTK is mathematically equivalent to linear regression on a fixed set of feature functions corresponding to the components of the Jacobian at initialization. The complement to this is sometimes called the 'feature learning' regime.

[22]For example, 'field' in a usual field theoretic sense corresponds with a function on a continuous domain, but in lattice models physicists sometimes define a field on discrete sites. Similarly, AI researchers tend to use 'feature' haphazardly. These are two examples of concepts that can be murky for anyone without inside expertise.

cannot? We also aim to connect work being done from the physics community with empirical studies of kernel eigenmodes within AI safety, like influence functions(Kreer et al., 2025), providing a theoretical bridge.

There are many axes along which we can categorize model complexity and behavior. We find it useful to think of the literature according to the following descriptors, noting that this is a partial list and that many references will cut across them:

- Whether it can be modeled by a static (equilibrium) or dynamic (training-time dependent) statistical theory. This often aligns with a specific asymptotic kernel (NNGP covariance vs. NTK), and dictates which observables are natural and how we interpret relevance.

- Whether training is well-approximated by linearized dynamics (lazy/kernel learning regime) or has substantial kernel drift (feature learning regime).

- Whether the theory is free or interacting. In parameter regimes where the 1/width (or another appropriate scaling parameter) is not small, the second order term is no longer sufficient to tell the whole story. This is important for understanding to what degree existing theory approximates real-world behavior.

- What the background is. A non-zero mean effectively changes which kernel modes couple to one another, impacting our choice of cutoff and determination of feature relevance. In the dynamic case, the mean also evolves during training, leading to further kernel drift.

The physics literature primarily uses language from statistical and quantum field theory, which expands a field with many correlated degrees of freedom in order of its fluctuations (correlation functions, which encode the ways in which degrees of freedom can interact) around its background (or vacuum expectation) value. Importantly, the second order term in this expansion is the covariance matrix or propagator (the kernel) encoding quadratic interactions. In many cases, this expansion is controlled when the width is large (in the limit, infinite), which makes higher order interactions subleading. This is often, though inaccurately, referred to in the ML literature as mean field scaling (or mean-field parametrization), which essentially limits the cumulant hierarchy to second order in 1/width (properly, the Gaussian or free field limit). Tools from perturbation theory and mean field theory can then be used to relax this limit, allowing one to capture finite-width behaviour of real-world models in a controlled manner (e.g., Roberts et al. (2022); Grosvenor & Jefferson (2022)). Other scalings (e.g., $\mu P$, adaptive or kernel scaling (Yang & Hu, 2022)) aim to capture the non-perturbative behavior by effectively removing the dependence on width. We point the interested reader to the review by Ringel et al. (2025b) of statistical physics applications to NNs for more information. While most of the work considered in this section comes from the physics community, we hope to connect it to related work from other fields. We think work on random matrix theory (e.g., Staats et al. (2025); Martin & Mahoney (2018)) will be a particular promising point of intersection.

### D.1.1 PHYSICS PERSPECTIVE: THE NTK

In Roberts et al. (2022) The Principles of Deep Learning Theory (PDLT), the authors derive neural tangent kernel regression as the leading approximation obtained by performing a linear expansion in the weights and accumulating the resulting first-order updates across gradient descent steps. In this regime, the NTK stays effectively fixed during training. This linearized approximation holds for models with a small depth-to-width aspect ratio and sufficiently many samples relative to the width, whose weights are initialized with a 1/sqrt(width) scaling. Finally, they extend the Taylor expansion to next-to-leading order in the weights and identify conditions under which a quadratic correction dominates higher-order terms. The resulting formula is well-approximated by kernel regression with the NTK at initialization replaced by the average of itself and the empirical NTK (eNTK). These results can be connected to renormalisation in two distinct ways:

- The kernel eigendirections define an intra-model RG. The NTK (or eNTK) and the function it approximates, can be further approximated by truncating eigendirections whose eigenvalues are below a certain cutoff. For different cutoffs, this yields a sequence of increasingly coarse-grained approximations to the function learned by the NN, via kernel regression with the truncated kernel. Such truncation can be a natural thing to do when the kernel

spectrum itself is highly anisotropic, with large fall-offs in its ordered eigenvalues setting effective emergent scales.

- Alternatively, from the point of view of an inter-model RG in a space of hyperparameters (width, depth, scalings of weight initializations with respect to them ... ), the NTK corresponds to a free universality class. During training, the same (data, training task) tuple can move along different trajectories (parameterizations) in model space depending on their hyperparameters and initializations, ending in a basin of attraction that characterizes its behavior. In the regime discussed here, which we can call the NTK basin of attraction, the models exhibiting no representation learning in the sense that they are nearly mathematically equivalent at the end of training to linear regression with a large number of fixed feature functions.

The authors of PDLT also invoke a third RG analogy in their book, at a structural (i.e., static) level. In early chapters, they study how, at initialization, layer-wise statistics within a given model (e.g., correlations of neural activations across different inputs) evolve with layer depth. For different fixed activation functions, they show that different choices of the c-number coefficients for the weight and bias variances at initialization separate an "ordered phase" where different model inputs become indistinguishable at late layers, from a "chaotic phase" where small input differences rapidly decorrelate with model depth. In practice one would like to tune these coefficients to lie on a critical line/manifold between the phases, so that inputs can propagate to the output layer in a reasonable way. In short, this version of the RG analogy guides select hyperparameter choices to avoid the problem of vanishing/exploding gradients.

Recent work suggests that this perspective is relevant for interpretability. Borrowing from statistical mechanics, where susceptibilities measure an observable's linear reponse to external perturbations, the authors of Baker et al. (2025) develop susceptibility-style probes to understand how observables localized on individual model components (for example, the per-component loss) respond to infinitessimal perturbations of the data distribution. This framework is conceptually close to probing sensitivity of kernel eigenmodes, and offers a principled way to quantify relevance at a scale defined by the chosen perturbation: components with high susceptibility contribute disproportionately to observable behavior, while those with low susceptibility can be considered irrelevant for those distributional shifts.

Complementary to this, Kreer et al. (2025) develop Bayesian influence functions that provide a Hessian-free (i.e., more scalable) approach to data attribution. Influence functions measure how much a particular training example affects model predictions, which naturally aligns with the eNTK picture at finite width. In an RG sense, this work offers a tool for identifying which data points are 'relevant' in the sense that their removal would significantly alter the effective description of the learned function.

### D.1.2 PHYSICS PERSPECTIVE: FIELD-THEORETICAL APPROACHES TO NETWORK STRUCTURE (NN-QFT)

'Neural Networks and Quantum Field Theory' introduces one of the earliest frameworks in "NN-QFT" or "NNFT" correspondence Halverson et al. (2021). Based on a mapping between NN architectures, whether at initialization or at any time-step during training, and statistical field theories (an Euclidean or thermal version of quantum field theories – the backbone of particle physics and string theory), this work primarily builds on field theoretic interpretability of NNs and data. It connects to both Grosvenor & Jefferson (2022) and Grosvenor & Jefferson (2022), in leveraging theoretical physics for new tools in mechanistic interpretability.

NNFT correspondence models any NN's output functional distribution as a statistical field theory "action" (model log-likelihood analog). For NNGP, the action contains a single term quadratic in the model output, which is generally diagonalizable on the basis of NN inputs (features); to borrow theoretical physics jargon – "the NNGP action is local in input space". Deviations from NNGP introduce additional terms in the model action that are higher-than-quadratic order polynomials in output, known as "interactions", leading to a series sum with infinite such corrections parametrized by $1/N$, where $N$ is the width of the hidden layer(s). Independently and identically distributed (i.i.d.) parameters, e.g., at initialization or an NNGP trained via stochastic gradient descent with $L_2$ loss, constrain the quadratic term in the action as $N$-independent, while terms higher-than-quadratic

order are accompanied by increasing powers of $1/N$; as a concrete example, a $r$th polynomial of output in action is accompanied with a prefactor $1/N^{r/2-1}$. Non-i.i.d. parameters, such as feature learning limits, introduce mixed scalings of $1/N$ and $\vec{\alpha}$ in action, where $\vec{\alpha}$ induces statistical correlations among model parameters. The $1/N$ scaling in action offers a natural hierarchy over non-local mixings in NN input (feature) space, as different polynomial-order interactions in action increasingly capture the interplays of model outputs as functions of data at varying spatial or information-geometric separations.

This work further introduces a Wilsonian RG scheme over the space of model inputs (features or frequency or momenta, if Fourier-transformed): via an explicit cutoff $\Lambda$ on the data space. $\Lambda$ may be determined by finite dataset sizes or resolution scales: either case leads to approximations of the model log-likelihood. NNFT correspondence generates NN outputs as field configurations – any loss in input-level information transforms the field interactions systematically. More specifically, the shift from infinite to finite data is analogous to coarse-graining over low frequency (momentum) field modes, whereas changes in resolution scale act as coarse-graining over high frequency (momentum) modes, respectively framed as RG of IR and UV field theories. The first case leads to an "effective action" oblivious to explicit roles of long-range data (feature) interactions, whereas the latter case leads to an effective action oblivious to explicit short-range (UV) interactions. This work builds on the IR case, where coarse-grained data modes are originally incapable of deprecating dominant statistical moments of the model output distribution; therefore, RG simplifies model analysis while retaining output quality and safety metrics. Qualitatively, data-related approximations are propagated into analytically tractable RG flows and $\beta$-functions of field "couplings" in interaction terms of action. For example, infinitesimal transformations of the dataset boundary lead to infinitesimal shifts in coupling, creating a continuous flow with potential critical points that may indicate phase transitions in model output space. More broadly, keeping all model parameters and hyperparameters unchanged, any coarse-graining over dataset boundaries or resolution scales show up as rescalings of model action; NNs at large width may have interactions in the action that become increasingly relevant, irrelevant, or stay unchanged, in response to transformations in dataset size, boundaries, and resolution scales.

'The edge of chaos: quantum field theory and deep neural networks' (Grosvenor & Jefferson, 2022) is the second of two primary directions that goes under the name "NN-QFT correspondence". At the most basic level, the most obvious difference in these directions is that this approach is concerned solely with the structural properties of network internals, and explicitly constructs the dual statistical or quantum field theory; in contrast, Halverson et al. (2021) applies to the network output (treated as a functional), and writes down a model action with suitable $1/N$ couplings to match observed statistics. At the level of RG analogies, this approach is closely related to that of Roberts et al. (2022) in that it describes layerwise statistics in the large-but-finite-width limit, in which the ratio of depth to width plays the roll of the perturbative parameter controlling the cumulant expansion. Notably however, while the authors of Roberts et al. (2022) were careful to avoid the use of the term "field" due to the lack of any meaningful notion of distance within a given network layer, this approach obtains a bona fide field theory by taking the limit in the depth, where distance is meaningfully defined. This is physically interesting because it reveals a close parallel between deep (as well as recurrent) neural networks and well-studied systems in physics known as large-$N$ vector models. Practically, it is interesting because it allows a controlled computation of finite-width corrections relevant for real-world models via a rigorous application of perturbation theory (in contrast, Halverson et al. (2021) takes a more phenomenological approach to modelling the output functional (which plays the role of the field variable there), while Roberts et al. (2022) relies on a layer-by-layer coarse-graining procedure to obtain truncated layer statistics; here, a path integral is constructed directly from the structure equation of the network). In short, this allows the authors to compute the cumulants describing fluctuations, i.e., finite-width effects, in internal network statistics.

On one hand, this has advantages in that it provides a fine-grained description of network internals at initialisation that allows, e.g., computation of finite-width corrections to the critical regime at which network trainability is optimised. More abstractly, from the perspective of physics, it is promising because it provides a mathematical duality between the structure of deep neural networks and a statistical field theory in $0 + 1$-dimensions, and hence can be used as a starting ground for more principled studies of renormalisation, universality classes, and critical behaviour in this context. As it provides a fine-grained path integral for the network, it also allows for the explicit computation of interneuron correlation functions (i.e., observables). On the other hand, it has the disadvantage

of being agnostic about the output of the network itself, and current versions are purely static: they describe networks at initialisation and do not account for the evolution of trainable parameters under stochastic gradient descent. Additionally, it is unclear how to immediately generalise the approach (which has been developed only for vanilla DNNs and RNNs) to SOTA models such as transformers. More work is needed to cross these gaps.

'Wilsonian renormalisation of Neural Network Gaussian Processes' (Howard et al., 2025a) applies RG to the Gaussian Process (GP) kernel, systematically coarse-graining over unlearnable modes to obtain a flow of the ridge parameter (i.e., the variance on the observed noise on the regression target) in which the data sets the cutoff scale. Here, learnability is defined relative to the ratio of the ridge parameter over the average number of datapoints: feature modes are eigenfunctions of the GP kernel, with eigenvalues $\lambda$; when the latter is significantly smaller than this ratio, the modes effectively decouple from the learning process (they behave as though the noise were infinite, or equivalently the number of samples were zero). Thus, they can be integrated out to obtain an effective theory (in the Wilsonian sense) on the learnable (i.e., low-energy or IR) modes,[23] where each RG step consists of integrating out a momentum shell of higher feature modes to obtain an infinitesimal change to the ridge parameter. The data sets the cutoff scale in the sense that this process naturally halts at the first learnable mode, whereupon we can relate the bare (unrenormalized, starting) ridge parameter to the effective ridge parameter that more accurately describes the Gaussian Process.

At a practical level, the authors show that this approach can predict the power-law scaling of the mean-squared error (MSE) loss obtained with many real-world data sets. More abstractly however, this is appealing because it goes beyond earlier structural analogies between NNs and RG, and establishes a practical link between RG and learning. In doing so, it provides a concrete stepping-stone towards establishing notions of universality in deep learning.

### D.1.3   PHYSICS PERSPECTIVE: CONNECTING DIFFERENT KERNEL REGIMES

A growing body of work aims to unify the lazy and feature-learning regimes for a more complete picture of how kernels evolve during training. 'Mixed Dynamics In Linear Networks: Unifying the Lazy and Active Regimes' (Tu et al., 2024) show theoretically that for a two-layer linear network $A = W_2 W_1$, one can derive an approximation to how the model changes under each step of gradient descent that contains lazy and feature-learning dynamics in different limits of the weight initializations and the model width. Their formula reveals a mixed regime where some singular values of A evolve lazily, while others are active. The authors also empirically locate a mixed regime in a phase diagram for a noisy matrix reconstruction task. This suggests that, in the context of the "inter-model" RG picture described above, it may be more natural to distinguish between lazy and feature-learning basins at the level of individual singular values of a matrix model (or in nonlinear networks, individual kernel eigendirections), instead of only at a model-wide level.

Naveh & Ringel (2021) extends the NNGP correspondence to the feature-learning regime by retaining non-Gaussian statistics at finite-width in two-layer CNNs. Feature learning effects emerge through a shift of the GP's target function; their theoretical framework involves solving a nonlinear self-consistency equation for higher-order cumulants of the network, which act as effective couplings beyond the GP. Empirically, they validate the approach for a linear network in a teacher-student setup, including evidence of a phase transition between lazy and feature learning regimes. In this example, the second-order NNGP kernel at initialization is the leading term in an effective description and does not change structurally during training, with feature learning effects entering as a data-dependent shift of the effective regression target (a non-centered GP prior).

A complementary line of work makes the kernel itself a dynamical object during training. In Bordelon & Pehlevan (2022), the authors develop a dynamical mean-field theory (DMFT) to describe feature learning in infinite-width networks, where interactions between layers are decoupled. Unlike the NTK/kernel regression picture, this formalism implements full gradient-descent parameterized by a feature-learning scale that smoothly interpolates between lazy and rich regimes in this limit. In this setting, the kernel is not just an initialization statistic, but a generator of order parameters – inner products of layer-dependent activations and gradients at pairs of time points. Kernel evo-

---

[23]Note that the role of the IR here is the opposite of that in Berman et al. (2023) mentioned elsewhere in this review, which considers a complementary application of RG in which the IR theory consists of a random DNN with unlearnable modes.

lution is governed by self-consistent saddle-point equations (the DMFT equations), resulting in an effective flow, with the fixed point given by a non-trivial function of hyperparameters. The same authors extend this framework to include finite-width fluctuations over random initializations that are non-perturbative in the feature-learning scale Bordelon & Pehlevan (2023). They find kernels are effectively static in the lazy limit, while in rich regimes kernels and prediction fluctuations become dynamically coupled, with a variance governed by the DMFT. They use this framework to analyze when and how width, learning rate (including edge-of-chaos phenomena), depth, training set size, and linearity, control the onset and endpoint of feature learning in both toy settings and CIFAR-10.

Fischer et al. (2024) connect feature learning to classical edge-of-chaos theory in deep nonlinear networks. They show the Bayesian prior can be written as an ensemble of GPs, and interpret feature learning as a reweighting of the components based on training data, framing kernel adaptation as arising from fluctuations in the prior. The capacity for adaptation (i.e., the flexibility afforded by these fluctuations) is maximized at the critical point separating ordered and chaotic phases, where the relevant response functions become large. This provides a theoretical link between criticality, response functions, and feature scale: networks initialized near criticality have the strongest capacity for kernel adaptation. Interpreting this through our lens, it suggests that feature relevance is sensitive to scale and the model's effective regime (IR basin) determined by training.

### D.1.4 PHYSICS PERSPECTIVE: SCALING LAWS AND PHASE TRANSITIONS

A useful distinction when conceptualizing feature learning is between *kernel rescaling* – where training changes the overall magnitude of the kernel from initialization, leading to GP-like generalization – and *kernel adaptation* – where training induces directional, data-dependent changes to the effective kernel. This rescaling v. adaptive axis asks 'what changes about the kernel?' during training. A second, largely orthogonal axis concerns different scaling regimes, in particular mean-field scaling (saddle-point approximation suffices) and standard scaling (where addition corrections are required). Rubin et al. (2025b) bridge these perspectives by rewriting the posterior over network outputs as a variational problem, and show that different choices of the dimensionality – an order parameter of this problem – recover either rescaling (low dimensionality) or adaptive (high dimensionality) theories. Separately, a systematic expansion of the output distribution adds the second axis to this picture. For the mean network output in the special case of linear networks, they find that kernel adaptation can be reduced to an effective rescaling—explaining why some feature-learning phenomena do not appear in rescaling-only theories. However, even in this case the framework captures adapative effects (i.e., of the output covariance) that cannot be captured by rescaling alone. This distinction aligns with our agenda: rescaling corresponds to a change in the scale of effective description, while adaptation corresponds to a change in its content.

As evidenced by the many approaches in this review, the abundance of factors that contribute to learning make our efforts to develop a comprehensive theory exceedingly difficult. In a renormalisation framing, this 'curse of detail' makes defining a single coarse-graining scale similarly hard to do. Through the lens of sample complexity, Rubin et al. (2025a) propose heuristic scaling arguments for predicting when different patterns of feature learning emerge. Rather than solving full DMFT equations numerically, they develop dimensional arguments that reproduce known scaling exponents and extend to complex architectures including three-layer nonlinear networks and attention heads. This provides estimates for the data and width scales at which transitions between lazy and feature-learning regimes occur—directly relevant to sample complexity in interpretability. If a task requires a threshold amount of data before features become relevant, interpretability tools assuming stable features may fail in the low-data regime. In Bordelon et al. (2024), the authors develop a solvable random-feature model to describe various scaling laws – in time, model size, and dataset size – and their various regimes of validity[24]. Relating to earlier work, they find a DMFT description of the large-width asymptotics, and solve the corresponding response functions exactly to find that the loss scales more quickly in larger models. For power-law distributed features, they predict that performance scaling with training time and model size follows different power-law exponents, leading to an asymmetric strategy for achieving compute-optimality. Importantly, they empirically verify that while random feature (fixed-kernel) networks follow linearized scaling laws, they fall short of the compute-optimal learning curves set by feature learning networks, suggesting that true understanding of this frontier will depend on a more 'mechanistic theory of feature learning.' Finally,

---

[24]We list just a few of these here; an expanded list can be found in the paper.

Rubin et al. (2024) apply an adaptive kernel approach to study grokking to understand if this phenomenon – a sudden increase in test accuracy before generalization emerges – is really out of reach of lazy/ GP theories of learning. Analyzing teacher-student models on two tasks (cubic-polynomial and modular-addition), they demonstrate a mapping between grokking and the theory of first-order phase transitions. Before grokking, the network is well-described by Gaussian feature learning, marked by the smooth adaptation of pre-activation covariances to the target directions in which their fluctuations – though peaked in the relevant directions – remain Gaussian. During grokking, features emerge discontinuously, yielding a mixed phase with pre-activation statistics described by a mixture of Gaussians. After the transition, the latent kernels develop entirely new features aligned with the teacher that alter sample complexity relative to GP limits.

### D.1.5 A Mixed Persepctive: Saddle-to-saddle dynamics, different kinds of emergence, and the ultra-rich regime

One can roughly trace two (non-mutually-exclusive) views of the emergence of structure in complex systems: the thermodynamic approach and the compositional, or mechanism-based, approach. This distinction is most famously made in Anderson's "More is Different" Anderson (1972). A more precise version is formalized in Rosas et al. (2019). The thermodynamic picture studies emergent macroscopic order in high-dimensional systems, usually viewed as composed of many identical microscopic particles (e.g., neurons in an ML context). Structure often appears at a phase transition and as emergent macroscopic order arising from complex microscopic interactions.

The mechanism-based approach instead focuses on highly symmetric or ordered systems where low-dimensional pieces emerge locally (often via symmetry-breaking along single neurons or linear directions) and build up precise mechanistic structures, like clockwork components combining into a timepiece.

Most of the work that has been presented here so far is in line with the thermodynamic picture. The mechanism-based view is perhaps a useful frame on the ideal *output* of an interpretation, perhaps obtained by extracting the relevant high-level mechanism after integrating out microscopic structure. In many idealized or highly symmetric settings, learning can even be fully understood in a mechanism-based setting. This framing is particularly suited to interpreting deep linear networks in the *ultra-rich* learning regime, and the related "saddle-to-saddle" view of learning dynamics.

The ultra-rich regime corresponds to initialization and target scaling that starts close to an unstable saddle-point where all weights are zero, and early learning proceeds by symmetry breaking (neurons or features "rolling away" from the unstable equilibrium). This dynamical picture is neatly captured in Kunin et al. (2025) on alternating gradient flows. It finds that in a modular addition model with small initializations, single neurons spontaneously "jump" from being close to zero to emerging as macroscopic pieces of the known mechanism for learning the modular addition task (a variant of Nanda et al. (2023a)) associated with Fourier modes. These large-scale neuron jumps alternate via "fast" but microscopic lazy learning-flavored re-adjustments of the small neurons as they react to a changing macroscopic target. In the weight landscape, this corresponds to the macroscopic parameters jumping between unstable saddles with better and better loss, with microscopic parameters quickly adjusting to metastable minima.

Saxe et al. (2013), and a cluster of related work, considers a similar phenomenon in the deep linear setting. Here a very small initialization in the ultra-rich regime bakes in two kinds of symmetry:

1. Localization to the task-aligned manifold (stable in idealized or simple settings). The training curve at each layer mostly stays within a lower-dimensional space of *target-aligned* matrices which are diagonal in the singular value basis of the target (a linear subspace of weight space in the case of deep linear models).

2. Scale degeneracy (unstable and trained away). At initialization, the singular values of the weight matrices are approximately zero (if they were exactly zero, this would be an unstable fixed point).

In Saxe et al. (2013), the approximate localization symmetry is reinforced (preserved) over training, while scale degeneracy gets broken as singular values learn the directions of the target. The paper derives a formal model for this that also predicts, and empirically confirms, the exact rate of learning

of each singular value. They find that large singular values get learned early and small singular values get learned late.

Dominé et al. (2024) studies the 2-layer linear network case in a more general context where the initialization is not assumed to be very small (i.e., outside the ultra-rich regime). By manually enforcing the basis-alignment property (item 1 above), they model the dependence of training curves on initialization in more detail and trace the difference between rich learning (where learning singular values exhibits symmetry breaking-like emergence) from lazy learning (where it does not, and learning is mostly described by adapting a single layer). They find that in addition to the initialization scale of each layer, an important parameter is the *difference* between initialization scales of the two layers (if one is too large, the asymmetry may result in only one layer effectively learning). This result partially extends to the case without basis-alignment, where exact learning dynamics is harder to predict but heuristically similar patterns hold.

In "Get Rich Quick" (Kunin et al., 2024), the exact shape of transition from lazy to rich learning is theoretically fleshed out for a tiny width-one setting (the opposite of the usual large-width picture), and a series of conservations laws that control the dynamics in this case are found. They find that nonlinearity and depth accelerate the lazy-to-rich transition.

The paper Chen et al. (2023) theoretically formalizes an analog of the property of "symmetry restoration" (item 1 above), where small initializations bias towards lower-dimensional symmetric "mechanism-like" solutions in simple tasks — in this case, the relevant property is equivalent to sparsity in the neuron basis. It finds evidence of this in training vision models.

Hoogland et al. (2025) finds evidence of approximate saddle-to-saddle structure in early learning of text transformers. The paper also tracks local thermodynamic properties of the loss landscape, and interprets them by analogy with the singular learning formalism of Watanabe (Watanabe, 2009). Singular learning formalism extends the study of low-dimensional symmetry breaking due to algebraic symmetry or Hessian degeneracy to a larger setting of geometric symmetry breaking due to analytic degeneracies of the loss landscape.

The saddle-to-saddle perspective on learning is most directly applicable in early learning and simple or low-dimensional settings. In larger and more complex settings, exact saddle-to-saddle phenomena (such as meachanistic explanations built out of single-neuron structures) become enmeshed with high-dimensional and noisy phenomena. For example, Kumar et al. (2023) studies modular addition in a rich (rather than ultra-rich) regime and a (roughly) mean-field setting. Here, the precision-tuned single-neuron jumps are replaced by high-dimensional statistical physics effects that explain grokking as a high-dimensional phase transition rather than a saddle-to-saddle phenomenon. Even settings that start ultra-rich can lead to more general settings with high-dimensional phenomena distinct from saddle-to-saddle replacing or interacting with the low-dimensional saddle-to-saddle structure. A few papers starting from the setting of Saxe et al. (2013) look at a learning process that starts in the ultra-rich regime but ends up in the mean-field or other regimes, usually in the context of task reversal and continual learning. For example Lee et al. Lee et al. (2021) studies a continual learning context where the target is explicitly changed during learning in a way that mimics implicit training realignment related to feature learning phenomena in large models. This paper finds exact ODE descriptions of the resulting non-saddlepoint regime with new phenomena (like non-monotonic forgetting).

Despite the fact that the pure saddle-to-saddle picture of emergent low-dimensional transitions is insufficient to describe complex NN learning completely, it provides a fully-tractable platform for understanding systems with significant complexity and interesting learning behavior. It is likely that experiments and intuitions from this simplified setting extend to nontrivial insights about late-training and non-ultra-rich settings, especially when studied at appropriate scales and renormalisation settings.

## D.2 DATA-SPACE RENORMALISATION

Much of the renormalisation idea hinges on how a NN represents the rich structure of datasets, from coarse regularities (long-range correlations) to fine idiosyncrasies (short-range). Data-space approaches consider:

1. How structure should inform our understanding of model-natural scales and coarse-graining schemes and the closure they afford at each scale (e.g., statistical, informational, causal, computational).

2. How renormalisation-like schemes map onto the way data features are compressed.

3. How compressed representations can be reliably interpreted up to a scale of abstraction resonant with the data structure.

Although they are less developed than kernel methods, data-space approaches may have more direct engineering applications. We expect synthetic data models with a known hierarchical ground truth to be particularly important in formulating a framework for both implicit and explicit renormalisation; capable of testing theoretical hypotheses and benchmarking new tools (like, e.g., Matryoshka SAEs (Bussmann et al., 2025)). Future work could also connect kernel renormalisation to data-space renormalisation, for example RG-based approaches like real-space mutual information (RSMI) (Koch-Janusz & Ringel, 2018)) and causal states inference (Shai et al., 2025; Rosas et al., 2024a).

Work in this direction does not cleanly separate by discipline. Instead, we consider that data can possess at least three categories of hierarchical structure. This structure may exist among the input features, on the data distribution as a whole, or within the target function. With input-space hierarchical structure, each input is iteratively decomposable into parts and subparts that represent increasingly fine-grained details. In this case, a hierarchical compositional relationship exists among the features or dimensions of each sampled input. With data-space hierarchical structure, the data distribution is iteratively decomposable into increasingly fine-grained subdistributions or clusters of data points. In this case, the hierarchical relationship exists among groups of data points. Finally, for data with functional hierarchical structure, a hierarchical relationship exists among the target function's components, such as terms in its series expansion. A given dataset may possess all, some, or none of these three complementary hierarchical structures.

### D.2.1 A MIXED PERSPECTIVE: HIERARCHICAL STRUCTURE IN INPUT SPACE

Hierarchical structure in input space can have different interpretations, depending on the modality. In general, classification tasks in which the input features are related by a regular compositional structure can be efficiently learned by deep neural networks (Mossel, 2018; Malach & Shalev-Shwartz, 2018). Several recent works analyzing the capabilities of transformer-based language models use a probabilistic context-free grammar (PCFG) as a synthetic model of hierarchical structure present in natural language (Allen-Zhu & Li, 2025; Menon et al., 2025; Lubana et al., 2024). A PCFG consists of a set of terminal symbols, a set of nonterminal symbols including a start symbol, and a set of probabilistically weighted production rules transforming a nonterminal symbol into a string of terminal and/or nonterminal symbols. A string generated by recursively applying a PCFG's production rules has a hierarchical structure that imposes long-range correlations among the observed terminal symbols or tokens.

A particular PCFG-based data model that has been applied in both language and image contexts is the random hierarchy model (RHM) (Cagnetta et al., 2024). The RHM is defined to have a regular structure and no ambiguities between the possible production rules that can yield an allowed string, making it an especially tractable model of hierarchical compositional data. A sequence generated by the RHM is represented by an s-regular syntax tree with $L$ levels. Each level has a distinct set of v possible nonterminal symbols, or terminal symbols for the final level. There are $m$ randomly chosen and frozen production rules per nonterminal symbol, which are constrained to generate unambiguous productions. In the basic RHM, all production rules are weighted with uniform probability. The RHM can be applied as a data model for classification tasks by predicting the highest-level latent (nonterminal) symbol as a label, or for next-token prediction tasks by masking and predicting the rightmost observable (terminal) symbol.

The RHM has been further studied in a number of follow-up works. In Cagnetta & Wyart (2024) is acquired by deep neural networks, the authors analyze token-token correlations in the RHM, showing that a finite training set has the effect of imposing an effective context window that limits the range of correlations to which the learned model is sensitive. Because learning additional correlations reduces the loss, leading to a power-law scaling law for next-token prediction. In Cagnetta et al. (2025), the analysis of scaling laws is extended to the case in which the production rules have a

power-law distribution at one level of the hierarchy. In this case, power-law scaling occurs for classification, while the exponent for next-token prediction is unchanged. In Sclocchi et al. (2024; 2025), the RHM is employed as a synthetic data model to analyze diffusion for both image and text data. By analyzing the RHM, the authors predict and verify on real data a phase transition in forward-backward diffusion experiments, where at a critical time scale the probability of reconstructing the true class drops to 0.

Another data model with hierarchical input-space structure that has been applied for image data is an iterated function system (IFS) that generates a self-similar fractal. IFS fractals have been used to generate synthetic data for pretraining image models (Nakamura et al., 2024; Baradad et al., 2022; Kataoka et al., 2021). The self-similarity of IFS fractals as a model for image data is suggested by the fractal appearance of natural structures produced by physical processes (Mandelbrot, 1983).

### D.2.2 A MIXED PERSPECTIVE: HIERARCHICAL STRUCTURE IN DATA SPACE

In comparison to input-space structure, hierarchical structure in data space appears to be less studied. One approach is to use an IFS fractal model of the data distribution rather than of input images. Machine Learning and Fractal Geometry reviews various approaches for developing classical machine learning algorithms that incorporate an inductive bias for fitting data distributions with fractal geometry. Bloem & de Rooij (2017) presents an expectation-maximization algorithm for fitting an IFS fractal model to data. Malach & Shalev-Shwartz (2019) shows that data distributions with fractal structure can be expressed efficiently by deep networks, but not with shallow ones. However, to learn a classification task on such a distribution using either a deep or shallow network, the negative examples must be coarsely distributed rather than concentrated on the distribution's fine structure. In addition, fractal-based methods are a well-studied technique for estimating the intrinsic dimension of data (Grassberger & Procaccia, 1983).

Another approach that models a data distribution as a stochastically self-similar fractal is based on percolation theory on a hypercubic lattice (Brill, 2024; 2025b). In this model, the data distribution consists either of discrete clusters with a power-law size distribution or a single dominant cluster, leading to a prediction of regime-dependent power-law neural scaling laws. Its self-similar fractal geometry suggests that data distributions can be efficiently mapped by learning sparse context features that hierarchically identify the clusters and subclusters to which inputs belong. Brill (2025a) applies this data model to study scaling laws for a capacity-constrained predictor that balances between task-specific or general-purpose capabilities, finding that general capabilities emerge abruptly, before declining in relative importance.

### D.2.3 A MIXED PERSPECTIVE: HIERARCHICAL STRUCTURE IN TARGET FUNCTIONS

Finally, target functions associated with natural learning tasks may be constrained to have hierarchical structure. Functions with sparse compositional structure, consisting of a hierarchy of constituent functions that each depend only on a small number of input variables, can be expressed efficiently by deep but not shallow neural networks (Poggio et al., 2017). Because any function that is efficiently Turing-computable is compositionally sparse, Danhofer et al. (2025) argue that the ability to exploit compositional sparsity is essential to deep learning's success. One possible reason that natural target functions may be compositionally sparse is the hierarchical structure of physical generative processes that are characterized by local interactions between levels in a sequence of scales (Lin et al., 2017). However, while deep networks can efficiently represent compositionally sparse functions, it does not guarantee that they can efficiently learn them. One function class that has been proposed as a model of functions that deep neural networks can efficiently learn hierarchically are staircase functions, which have a compositional structure in which high-order Fourier coefficients are built up from lower-order coefficients step by step (Abbe et al., 2021; 2023).

### D.2.4 A PHYSICS PERSPECTIVE: REAL-SPACE RG AND INFORMATION-THEORETIC COARSE-GRAINING

One of the first works to relate deep learning and renormalisation was by Mehta and Schwab (Mehta & Schwab, 2014). The authors argued that restricted Boltzmann machines (RBM) perform implicit renormalisation that is analogous to Kadanoff's variational renormalisation scheme from condensed matter physics (Kadanoff et al., 1976). RBMs are unsupervised energy-based models that can learn

high-dimensional probability distributions, and they are closely related to equilibrium spin models. However, later work suggested that, without additional assumptions, representations uncovered by RBMs do not always recover the types of large-scale features that are usually sought in renormalisation methods (Lin et al., 2017; Koch-Janusz & Ringel, 2018; Schwab & Mehta, 2016). We also note that RBMs were found to be difficult to scale, and so they are not frequently used in modern ML systems.

A different line of work has employed ideas from machine learning and information theory to develop explicit renormalisation methods. In the physics literature, renormalisation methods often require system-specific knowledge (such as the order parameter or the symmetries of the Hamiltonian) and hand-coded renormalisation rules. Koch-Janusz & Ringel (2018) developed a more general approach which discovers such information and renormalisations from data, rather than requiring that it be provided a priori. Their so-called "real-space mutual information" (RSMI) method is well-suited to studying homogeneous spatial systems. In such systems, it considers a local region $V$ that is separated by a buffer region B from a distant environment region $E$. RSMI then identifies a lower-dimensional hidden variable $H$ that depends only on local region $V$ while also having large mutual information $I(H; E)$ with $E$. In simple terms, $H$ is the renormalized version of $V$ that encodes relevant information about large-scale features. More recently, the RSMI method was improved by using powerful ML techniques for quantifying mutual information, and it was demonstrated on various non-trivial physical systems (Gordon et al., 2021; Gökmen et al., 2021; Gökmen et al., 2021). It has also been shown that, by considering graph-based distances, RSMI can be extended beyond homogeneous spatial systems to inhomogeneous systems on a graph (Gökmen et al., 2024).

RSMI has two intrinsic notions of scale. The first notion is the size of the buffer region $B$, the non-predicted region that filters out short-range correlations, and thus sets the spatial scale of interest. The second notion is the dimensionality of the hidden variables $H$, which sets the degree of information compression during coarse-graining. Recent work has shown that information compression can also be controlled by minimizing the mutual information $I(H; V)$, rather than limiting the dimensionality of $H$ (Gordon et al., 2021), revealing an important formal connection to the "information bottleneck" method (Tishby & Zaslavsky, 2015; Kolchinsky et al., 2019; Saxe et al., 2018).

At a high level, RSMI seeks features that capture long-range spatial correlations in statistical ensembles. Conceptually, it is related to methods for coarse-graining of dynamical systems that seek features that capture long-range temporal correlations (Schmitt et al., 2025; Rosas et al., 2024b; Pfante et al., 2014; Shalizi & Moore, 2024; Görnerup & Jacobi, 2007). In fact, recent work has developed a coarse-graining approach to dynamical systems that is similar to RSMI and information bottleneck (Schmitt et al., 2025), and demonstrated that it can recover important temporal features in complex physical and biological systems .

Another formal approach to identify useful coarse-grainings has been introduced in Rosas et al. (2024a). Here, coarse-grainings are 'useful' to the degree they capture levels of description that are self-contained from an informational, causal, or computational perspective. Informational self-containment is operationalised in terms of information-theoretic terms: it corresponds to when a macro scale can optimally predict itself without considering information from finer scales. In contrast, causal and computational self-containement is formalized utilizing computational mechanics (Shalizi & Crutchfield, 2001) – a framework that combines principles from statistical physics and theoretical computer science to investigate pattern formation in time series data. Results have shown that these properties are pervasive in paradigmatic models of statistical physics and computational mechanics. Moreover, it has been found that symmetry (more specifically, dynamical equivariance) is a sufficient condition for the existence of such self-contained macro levels (Rosas, 2025). This work has also shown that familiar macroscopic quantities (such as magnetisation or particle count) correspond to informationally closed macro processes arising from underlying symmetries.

### D.2.5 A PHYSICS PERSEPECTIVE: STATISTICAL INFERENCE AS RENORMALISATION

Statistical Inference itself may be framed in terms of renormalisation. In Berman et al. (2023) the link between traditional statistical inference using Bayesian methods and the exact renormalisation group flow was demonstrated as follows. First, consider a small incremental flow of data into your model. As the relatively infinitesimal data comes into your model one piece at a time you can use Bayes's theorem to update your model parameters. Of course, since the amount of new data is small

as compared with the amount of previous data on which the prior is based the parameter updates will be similarly infinitesimal. This infinitesimal Bayesian update equation can then be recast in terms of a first order differential equation for the change in the model parameters as a function of the amount of data. This first order differential equation can be reinterpreted as an exact RG flow equation, describing how physical parameters change with "scale".

In fact, we can read off two relevant scales that appear as a ratio describing how model parameters flow. First there is a scale associated with the model itself. This is the Fisher information metric for the model parameters. The Fisher information in some sense measures the distance in model space and so its appearance as a scale associated with the model is perfectly natural. Second, there is the scale simply given by the amount of data. Again this is natural and intuitive, the main scale on which all learning depends is data quantity. What is useful about the quantitative relationship that is described from the continuous Bayesian updating is that it provides a detailed description of the interplay between scales in the model given by the Fisher information and scales in data. Beyond the mathematics of the RG flow equation and its equivalence to the Bayesian updating equation, the basic intuition that this work demonstrates is that in traditional RG flow, one is throwing away information through "integrating out"; in statistical inference one is doing the opposite, using Bayes's theorem to add information to your model from new data. The fact that one is sort of the inverse of the other is reflected in the fact that when mapping the theories the physical scale in the RG equation becomes mapped to the inverse of the data quantity in the Bayesian update equation. Thus, large data corresponds to the ultraviolet and small data the infrared. This is of course natural from the physics perspective, as we observe more about the universe we can distinguish between different possible UV theories. Data provides the information that allows the inverse of traditional RG to flow from IR to UV.

Well observed phenomena are reproduced such as the variance in the statistical model as a function of data quantity (for small amounts of data, model variance is high, for large amounts of data the model asymptotes according to the central limit theorem). This description also allows a principled approach to model parameter pruning. One can implement a Bayesian renormalisation flow: progressively discarding parameter directions that the data cannot resolve (i.e. 'sloppy' modes) while retaining the informative, resolvable ones. Fisher information, in this context, measures how sensitively a probability distribution responds to changes in a parameter — directions with high Fisher information covary strongly with the output of the model, while those with low Fisher information blur into statistical noise. In settings with an underlying physical cutoff, this recovers conventional renormalisation (Howard et al., 2025b), but in general it provides a principled compression/inference scheme that respects what data actually supports. In the context of neural networks, this perspective suggests that multi-scale properties of data are reflected in parameter directions with differing scales of Fisher information, so that coarse-graining by informational relevance naturally defines scales intrinsic to the model. It was further shown that this aligns with the information bottleneck and can be realized in toy neural-network pruning experiments, where parameters are hierarchically organized by their informational significance. The technique has been demonstrated with a toy autoencoder, to perform systematic model pruning.

This can be thought of as a coarse-graining scheme over model parameters at each time-step of training in a way that preserves downstream performance metrics, via an information-geometric UV cutoff in parameter space. Features separated by distances smaller than a fixed resolution scale no longer contribute individually to the model's learning and inference processes; accordingly, the model log-likelihood is oblivious to explicit short-range data (feature) interactions. As a concrete example, information-theoretic RG over UV field modes in a model initialized at NNGP, trained via SGD at $L_2$ loss and leading to a different NNGP, simplifies to "mass renormalisation" of a dual free statistical field theory, leading to rescaling of parameter means and covariances. This framework does not require large hyperparameter or parameter assumptions that are traditionally required by methods such as random matrix theory or dynamical mean field theory. Alternatively, encoding data itself may be framed through the lens of field-theoretic renormalisation. It was found Berman et al. (2025) that one may design an encoder-decoder architecture whose latent space is explicitly composed of n-point correlation functions (or cumulants) of the input data. The NCoder treats images (or other high-dimensional data) as draws from a lattice field theory, then reconstructs them via a perturbative expansion in correlation functions—analogous to building an effective action order by order in quantum field theory. In doing so it establishes a correspondence between perturbative renormalizability and model sufficiency: if only a finite number of correlation functions are needed, the data

generating model is heuristically 'renormalizable'. The method was demonstrated on MNIST, with benchmarks showing that generated images can be classified correctly using only up to the 3-point functions of the latent summary statistics. This suggests, empirically, that in some sense MNIST data samples are perturbatively renormalizable. Thus one might view NCoder as implementing a renormalisation-like compression in statistical observable space (moments and cumulants) while Fisher renormalisation works in parameter space; comparing them could shed light on how models choose which scales of structure to maintain or discard.

### D.2.6 COMPRESSIBILITY AND INTERPRETABILITY

Similar work questions how different training dynamics encode features (Manning-Coe et al., 2025); it is known that when comparing grokking (i.e. sudden generalization after memorization) with steady training regimes, the same underlying features may be learned but are represented with markedly different efficiency and compressibility. In particular, a 'compressive regime' emerges in steady training where there is a linear trade-off between model loss and compressibility, a phenomenon absent in grokking. While not immediately framed directly in terms of Fisher information, this compressibility picture resonates with the Fisher perspective: parameter directions that carry little information are effectively suppressed in the compressive regime, whereas grokking does not enforce such selectivity. One may consider this akin to implicit renormalisation. In the present framework, this supports the idea that reliable interpretations of compressed representations should be limited to scales where features remain distinguishable in a Fisher-information sense, providing a natural boundary for abstraction within the data structure. The work also shows that models which learn through grokking are not necessarily more compressible in terms of Bayesian Renormalisation than those which learn through steady learning.

A related picture of compressibility finds direct support in recent work on sparse autoencoders. Matching pursuit SAEs Costa et al. (2025) replace the standard shallow encoder with a greedy iterative process: at each step, the algorithm selects whichever dictionary feature best explains the current residual, subtracts its contribution, and repeats. Features selected earlier in this process tend to be coarser-grained. Practically, this suggests neural networks may be able to perform a form of lazy evaluation over their feature hierarchy, first resolving coarse-grained concepts before committing computational resources to finer distinctions. Compressibility, in this sense, is the freedom to operate at coarse or fine resolution as needed.

Matryoshka SAEs (Bussmann et al., 2025) arrive at a similar conclusion via a different route. By training nested SAEs at progressively increasing widths, they find that capacity-constrained (narrow) models preferentially learn coarse features, with finer structure emerging only as width increases. That both methods recover the same coarse-to-fine structure strengthens the case that this organization is intrinsic to learned representations.

## E   GLOSSARY

### E.1   PHYSICS / RG TERMINOLOGY

**Coarse-graining:** Any map from a fine-grained description (microstates, parameters, features) to a coarser one, typically by aggregating or integrating out degrees of freedom.

**Correlation length:** A scale $(\xi)$ characterizing how quickly correlations decay in space or time. Finite $\xi$ typically implies exponential decay; $(\xi \to \infty)$ at critical points implies scale-free, power-law correlations.

**Critical exponent / scaling dimension:** An exponent characterizing how a quantity scales near a fixed point (e.g. with correlation length, temperature, or system size). At critical points, these are the usual critical exponents; more generally, they are eigenvalues/eigenvectors of the RG linearization.

**Critical point:** A point controlling a continuous phase transition, where correlation length becomes very large and many observables exhibit non-analytic, power-law behavior over many scales. Critical points are where universality is most dramatic.

**Effective field theory (EFT) / effective theory:** A theory valid only up to some scale, written in terms of degrees of freedom and interactions relevant at that scale. In this paper, "effective

theory" is used broadly for any coarser description of a model that preserves specified observables up to a tolerance.

**Fixed point (of the RG flow):** A point in theory space that is invariant under RG flow (up to rescaling). Near a fixed point, the theory often exhibits simple scaling structure.

**Hierarchical conditional independence (as used in this paper):** A Markov-like property in scale. For a good RG scheme, coarse variables at a given scale act as sufficient statistics for the finer variables directly beneath them, for the observables we care about. Conditioned on coarse variables, fine-scale degrees of freedom in one region are approximately independent of distant regions and have bounded influence on macroscopic observables (See Section D.1.5 for related formalizations using mutual information and causal states).

**Infrared (IR) / Ultraviolet (UV):** "IR" refers to large-distance / low-energy / coarse scales; "UV" refers to short-distance / high-energy / fine scales. In this paper, IR typically corresponds to scales where safety-relevant observables live; UV corresponds to microscopic parameters or very fine features.

**Locality (and quasi-locality):** The property that interactions in an effective theory involve only nearby degrees of freedom (or decay quickly with distance).

**Observable:** A measurable quantity or behavior of interest used to connect theory with experiment. In a NN, this could be loss on a task, performance on a benchmark, specific probe scores, statistics of internal activations, or response to interventions.

**Phase transition (continuous):** A non-analytic change in macroscopic behavior as parameters are varied, controlled by a critical fixed point. Often accompanied by divergent correlation length and scale-invariant fluctuations.

**Relevant / irrelevant / marginal (directions or operators):** A characterization of how linear perturbations around a point in theory space behave under an RG flow. *Relevant* directions grow under coarse-graining and strongly affect long-distance behavior. In an NN context, these are effective features whose perturbations significantly change chosen observables. *Irrelevant* directions shrink and become negligible in the IR. In an NN context, these form a tail with bounded impact. *Marginal* directions neither clearly grow nor shrink at linear order and require higher-order analysis.

**renormalisation (RG):** A procedure that relates microscopic descriptions of a system to effective descriptions at coarser scales, by integrating out fine-grained degrees of freedom and tracking how couplings change.

**RG flow:** The trajectory a theory traces in coupling space as we change the scale (under repeated coarse-graining and rescaling). Encodes how the effective description evolves from UV to IR.

**Scaling law:** A relationship showing how an observable changes under rescaling of length, energy, or other parameters—for example, power-law dependence near criticality. In this paper, we also use "scaling law" for empirical relations in NNs (e.g. loss vs. model size / data / compute).

**Separation of scales:** A property of a renormalisation scheme that allows fine-grained details to be integrated out without appreciably influencing coarse observables. Practically: a small set of effective degrees of freedom dominates chosen observables, and the contribution of discarded modes can be bounded.

**Theory space (or coupling space):** An abstract space whose coordinates are the couplings (parameters) of all operators allowed in a theory. RG flows are curves in this space; points represent effective theories.

**Universality:** The phenomenon where many microscopically different models flow under RG to the same IR fixed point and share the same long-distance behavior for a wide range of observables.

**Universality class:** A family of models whose RG flows end at the same fixed point and therefore share the same asymptotic large-scale behavior, often characterized by common scaling laws and exponents.

## E.2 INTERPRETABILITY TERMINOLOGY

**Causal mechanistic decomposition:** A way of representing a system as a hierarchy of interacting causal mechanisms such that the system's behavior factorizes into these modules while preserving key causal counterfactuals across levels of abstraction. We treat this as a target-like structure that a successful RG scheme would approximate across scales.

**Circuit:** A structured collection of internal components (neurons, heads, features, weights) whose combined activity implements a recognizable sub-computation (e.g., an induction head, an IOI circuit).

**Critical-like regime (in NNs):** A region of parameter or scale space where small changes in architecture, training, or data lead to sharp or qualitative changes in effective descriptions or capabilities (e.g., capability jumps, phase-like transitions in behavior). Potentially analogous to critical regions in statistical mechanics and useful as diagnostics.

**Effective feature / effective degree of freedom:** A coarse, possibly composite feature that emerges under a renormalisation scheme and captures most of the contribution to some observable at a given scale. The units in which we want to express the effective theory of a model.

**Explicit RG (tool):** A post-hoc procedure that takes a trained model or representation and constructs a coarser description (e.g., spectral truncation, clustering features, layerwise abstraction), aiming to mimic an RG step in a defined feature space.

**Feature (in this paper):** Any component that meaningfully represents model internals in a way that can be understood by a human. Examples include a direction in activation space, an eigenvector of a kernel or covariance operator, a sparse autoencoder (SAE) feature, or a component of a generative data model.

**Implicit RG (theory):** A theoretical description of how NN training or inference implicitly implements an RG-like organization of features.

**Mechanistic interpretability:** The process of understanding models in terms of circuits, features, and algorithms—identifying internal structures (e.g., attention heads, MLP neurons, feature combinations) that implement particular computations.

**Representation / internal representation:** The vector of activations (or a function thereof) at some layer or part of the network, taken as an internal state encoding information about inputs and context.

**Safety guarantee / bound:** A formal statement bounding the impact of a safety-critical behavior. In this draft, we consider these to be of the form: "conditional on this effective description, perturbations confined to the irrelevant subspace cannot change observable $X$ by more than $\varepsilon$." RG-inspired separation of scales is the structural property that would make such statements possible in restricted settings.

**Sparse autoencoder (SAE) feature:** A learned sparse basis function over activations, used to define interpretable features by reconstructing activations in a sparse code.

**Susceptibility (in NNs):** A measure of how sensitive a model, circuit, or feature is to a structured perturbation (e.g., feature ablation, parameter change, input intervention). Inspired by linear-response/susceptibility in physics; used here as a way to quantify relevance and bound influence.

**Universality-like behavior (in NNs):** Any regime where families of models (e.g., similar architectures and training recipes) share stable effective descriptions and scaling relationships for certain observables, even if not in the strong physics sense of universality classes at fixed points.