# OpenReview forum: "Towards Worst-Case Guarantees with Scale-Aware Interpretability"
_ICLR.cc/2026/Workshop/Sci4DL — Submitted to Sci4DL 2026_

### Official Review · Reviewer_fcT7 · 2026-02-12

**Fit:** 3
**Significance:** 3
**Confidence:** 2

**Summary:**

This position paper reviews applications of renormalizing group theory in neural networks and promotes scale-aware interpretability/safety research for the reason of worst-case guarantee (discarded fine-grained details *never* jailbreaks). It discusses the limitation of SAE lacking canonicity, the existing RG approaches including NTK/NNGP, NNQFT, grokking phase transition, and calls for a general renormalization tool.

**Strengths:**

- The discussed position is a long-lasting key issue in the domain of mathematics of DL.
- The topic is well-explained, well-motivated and the survey is thorough.
- The domain is perfectly aligned with the workshop.

**Suggestions:**

To broaden the horizon of the survey, I would like to suggest identifying additional field variables for the RG beyond NNGP/NTK (the “what” part in the paper):
- Weight eigenspace—RG gauge fixing applied to Muon: https://www.slideshare.net/slideshow/from-setol-to-muon-muon-is-rg-gauge-fixing/285123437
- Effective interaction scales in the universality class over attention (TMR/GRT): https://openreview.net/pdf?id=p0e8bS0ced
- There exist emergent applications of RG scaling on other dimensions recently: attention heads, ...

---

### Official Review · Reviewer_zaqj · 2026-02-26

**Fit:** 1
**Significance:** 1
**Confidence:** 3

**Summary:**

This paper proposes “Scale-Aware Interpretability,” a research agenda inspired by the renormalisation group from statistical physics. The authors argue that AI systems exhibit multi-scale structure and that interpretability methods should be tailored to such structure. They suggest that a renormalisation framework could enable tools that separate relevant from irrelevant features and potentially provide robustness and safety guarantees.

**Strengths:**

-The paper brings together ideas from statistical physics, kernel theory, feature learning, and mechanistic interpretability in a thoughtful way. The literature review is extensive and demonstrates deep engagement across multiple communities.

-The discussion of separation of scales, relevance, effective degrees of freedom, and hierarchical conditional independence provides a useful conceptual vocabulary for thinking about interpretability beyond linear feature decompositions.

**Suggestions:**

1. **Clarify the scope of “worst-case guarantees.”** The paper repeatedly suggests that renormalisation-inspired tools could enable worst-case guarantees, but this notion is not formally defined. As written, the claim feels aspirational rather than technically supported.

2. **Make the research agenda concrete.** While the agenda seems compelling at a high level, it remains too abstract. Providing even a minimal worked example would help ground the proposal and make the agenda more actionable. While a position paper is not inherently problematic for a workshop, the current writing lacks an indication of a) how the agenda proposed by the author could be implemented, b) what are the challenges of current interpretability methods that could be overcome with scale-aware methods.

3. **Tighten terminology around relevance and scale**. Although the glossary is helpful, several central notions (e.g., “model-natural scale,” “relevance,” and “separation of scales” in NN settings) remain highly metaphorical. Adding sharper definitions or toy examples would reduce ambiguity.

Overall, the paper presents a compelling and ambitious conceptual system, but would require a more concrete instantiation of the core methods and claims to fit within the workshop.

---

### Official Review · Reviewer_UDPW · 2026-02-26

**Fit:** 1
**Significance:** 2
**Confidence:** 1

**Summary:**

This is a proposition paper, with a call to action to apply methods and thinking from statistical physics (specifically the renormalization framework) to neural network interpretability. The belief is that this would allow one to filter out irrelevant variables with theoretical bounds on the error, and allow explanations at different resolutions. This approach is named "Scale-Aware-Interpretability".
The paper provides a brief overview of the normalization framework, before providing an overview of which questions such a framework should address for each interpretability use case, and culminates in a call-to-action to import certain ideas from statistical physics and apply them to neural networks.

**Strengths:**

The paper is well written, ambitious, and the proposed approach is also clearly framed within the context of existing literature. Furthermore, the resulting properties of solutions that could be found by applying this framework sound very compelling.

**Suggestions:**

The paper doesn't contain any experiments or clear examples of successfully applying this framework. It is also written in a general and broad fashion, therefore it is very unclear to me (unfamiliar with statistical physics) how it would be applied in practice, and I find myself unable to evaluate the feasibility and merit.

One concern is the fit of this work with the workshop agenda: while the proposition is certainly interesting, I don't believe it's very aligned with the spirit of the workshop, e.g. it doesn't exactly fit "In particular, we expect submissions to either conduct, rely on, or inform empirical experiments on real-world dataset" as outlined in the call for papers.

---

### Meta-Review · Area_Chair_DgYV · 2026-02-28

**Recommendation:** Reject

**Metareview:**

This paper seems to be a longer position paper that has been significantly shortened to fit into a workshop submission. As noted by two reviewers, this submission is not a good fit for the workshop, as it does not contain nor discuss experiments on real-world datasets. I therefore recommend rejection.

---

### Decision · Program_Chairs · 2026-03-02

Reject